# MoL: Adaptive Mixture-of-Length Reasoning for Efficient Question Answering with Context

**Guocong Li**[1,2*], **Jinjian Zhang**[2*], **Ping Wang**[4], **Dongnan Liu**[2], **Tian Liang**[5], **Qiuyi Qi**[5],
**Hao Huang**[2], **Siyan Guo**[2], **Mutian Bao**[5], **Wei Zhou**[2], **Linjian Mo**[2†], **Hongxia Xu**[3†], **Jian Wu**[6†]

[1]Transvascular Implantation Devices Research Institute and State Key Laboratory of
Transvascular Implantation Devices, Zhejiang University, Hangzhou, China
[2]Ant Group
[3]Liangzhu Laboratory and WeDoctor Cloud, Hangzhou, China
[4]Renmin University of China
[5]College of Computer Science and Technology, Zhejiang University
[6]Zhejiang Key Laboratory of Medical Imaging Artificial Intelligence, Hangzhu, China
{liguocong,einstein,wujian2000}@zju.edu.cn

## Abstract

We present Mixture-of-Length (MoL), an approach for Question Answering (QA) with context that aims to improve the balance between reasoning quality and response efficiency. Our method introduces a principled difficulty assessment based on information-theoretic principles and a dual-objective reward mechanism that adaptively modulates response length. In our experiments, MoL exhibits an emergent behavior termed "intelligent brevity": the model tends to produce shorter responses for simpler queries and longer ones for more complex inputs. This property is desirable for human-computer interaction and can reduce inference costs. A post-hoc analysis of internal activations suggests a correlation between this output adaptivity and the effective number of layers that contribute during inference. On multiple QA benchmarks, MoL demonstrates competitive accuracy while substantially reducing tokens compared to baselines, indicating that difficulty-aware length modulation is a promising direction for efficient QA with context. [1]

## 1 Introduction

Question Answering (QA) with context represents a fundamental challenge in natural language processing, where models must synthesize information from multiple sources to generate accurate responses. While recent advances in large language models (LLMs) have demonstrated remarkable capabilities in this domain (Suzgun et al., 2023), a critical tension persists between reasoning quality and computational efficiency (Pan et al., 2024; Su et al., 2024). This challenge is most acute in multi-document scenarios, where reasoning complexity varies from simple extraction to complex multi-hop inference. Efficiently navigating this spectrum is key. Figure 1 offers an initial glimpse of our solution on HotpotQA, demonstrating that comparable accuracy can be achieved with significantly shorter responses.

Current approaches to efficient reasoning fall into two primary categories, each with significant limitations. **First**, uniform compression methods (Yang et al., 2025; Kang et al., 2025) apply fixed reduction strategies regardless of task complexity, leading to under-reasoning on difficult problems while over-elaborating on simple ones. **Second**, adaptive methods (Ling et al., 2025; Team et al., 2025) attempt difficulty-aware processing but rely on heuristic difficulty estimation and rigid compression policies that struggle to recover when initial assessments prove inadequate.

---

* Equal contribution.
† Corresponding authors.

[1]Codes are available at: https://github.com/cong03/MoL.

The core insight driving our work is that optimal reasoning should be fundamentally adaptive, which involves allocating computational resources proportional to the inherent complexity of each query. However, realizing this vision requires addressing two key technical challenges: (1) principled difficulty assessment that can reliably distinguish between simple extraction tasks and complex reasoning problems, and (2) fault-tolerant adaptation that can dynamically expand reasoning when initial attempts prove insufficient.

We introduce **Mixture-of-Length (MoL)**, a novel framework that addresses these challenges through two key innovations. First, we develop a theoretically-grounded difficulty assessment based on information-theoretic principles, specifically modeling QA complexity through the lens of the Set Cover problem. Our metric quantifies reasoning difficulty by measuring cross-document information redundancy, where high redundancy indicates simple extraction tasks and low redundancy signals complex multi-hop reasoning requirements. Second, we propose a dual-objective reward mechanism that implements fault-tolerant adaptation: it encourages compression for correct responses while promoting expansion for incorrect ones, enabling the model to self-correct by scaling reasoning capacity on-demand.

Crucially, MoL exhibits an emergent behavior termed **"intelligent brevity"**: the model naturally learns to produce concise responses for simple queries and elaborate reasoning for complex problems, without explicit length constraints. This is a direct outcome of our training design, which encourages adaptive resource allocation based on question difficulty. Post-hoc analysis further reveals this output-level adaptation correlates with internal computational patterns: simpler questions activate fewer transformer layers, while complex ones engage deeper model capacity. This suggests that MoL induces a form of dynamic computational allocation operating coherently across both external responses and internal representations.

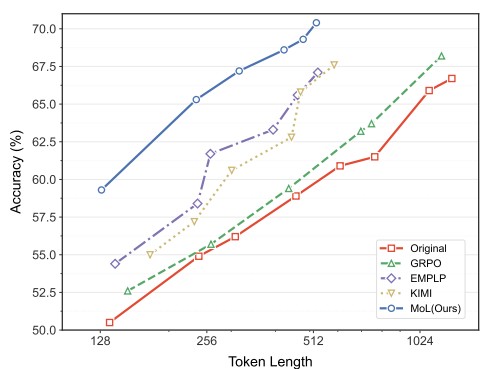

Figure 1: An evaluation of model accuracy and token length on HotpotQA: Original (base model), GRPO (RL with accuracy reward only), ER-PLP (difficulty-aware adaptive reasoning depth), KIMI (length-penalized reasoning compression), and MoL (ours). Lower tokens at similar or higher accuracy is better.

Empirical evaluations across multiple QA benchmarks, including HotpotQA (Yang et al., 2018), StrategyQA (Geva et al., 2021), and Loong (Wang et al., 2024), reveal that MoL significantly enhances both performance and efficiency. Our comprehensive results demonstrate that MoL achieves: a) Up to 53.2% reduction in token length in inference efficiency across QA with context datasets. b) An absolute accuracy improvement of 5.0%, outperforming state-of-the-art token compression and reinforcement learning methods. c) Superior performance generalization to unseen datasets, highlighting the robustness of our difficulty-aware approach.

Our contributions are threefold: (1) a principled, information-theoretic approach to difficulty assessment in multi-document QA, (2) a fault-tolerant adaptive reasoning framework that dynamically balances efficiency and accuracy, and (3) empirical evidence that output-level adaptation can reflect and potentially drive internal computational allocation in transformer models.

## 2 RELATED WORK

**QA with Context.** QA with context task aims to enhance question understanding by incorporating additional information, rather than treating questions as isolated inputs (Min et al., 2020). This context encompasses various factors, particularly cognitive and social factors related to user intent, tasks, and needs. Consequently, the effective utilization of context is crucial for accurately interpreting questions. Notably, the performance of Large Language Models (LLMs) remains challenged when handling long-text tasks due to limitations in their context window (Chiang & Cholak, 2022). Numerous studies are currently dedicated to extending the effective context length of LLMs (Xiao et al., 2024; Chevalier et al., 2023). Given this, it is essential to propose methods capable of effective task-specific adaptation for long-text tasks.

**Large Reasoning Model.** In recent years, large language models (LLMs) have achieved remarkable breakthroughs in complex reasoning tasks (Wei et al., 2022). A key innovation in this field is the Chain-of-Thought (CoT) technique, which enhances multi-step reasoning by introducing intermediate reasoning steps (Yao et al., 2023). This approach has significantly improved model performance in challenging tasks such as mathematical deduction and logical analysis. Building upon this direction, researchers have further integrated reinforcement learning techniques to enhance models' autonomous reasoning capabilities through answer feedback mechanisms (Cheng & Van Durme, 2024). This technical approach has led to several state-of-the-art models, such as OpenAI's o1 (Achiam et al., 2023) and DeepSeek-R1.

**Efficient Reasoning.** Several recent works have been proposed to address the redundancy in Chain-of-Thought (CoT) reasoning (Ma et al., 2025; Shen et al., 2025). For supervised fine-tuning, Yu et al. (2025) introduces the LS-Mixture framework, which mitigates redundancy by jointly training on both original long CoT sequences and their reconstructed shorter ones. Other methods enforce stricter constraints: Token-Budget imposes a hard token limit to streamline computation (Han et al., 2024), whereas TokenSkip adopts importance-weighted filtering (Xia et al., 2025). However, both approaches risk omitting pivotal reasoning steps, especially for complex problems. Reinforcement learning approaches have enabled more flexible optimization (Rafailov et al., 2023; Ferrag et al., 2025). The Kimi team's work utilizes a contrastive length reward to encourage conciseness (Team et al., 2025), and Ling et al. (2025) propose an adaptive strategy that adjusts reasoning depth based on pre-assessed problem difficulty. While effective, these methods often suffer from two key limitations: (1) they lack a mechanism for error recovery when an initially concise answer is incorrect, and (2) their compression strategies can be brittle, failing to expand reasoning for unexpectedly complex queries. In contrast, MoL introduces a fault-tolerant, dual-objective mechanism. It not only compresses responses for simple tasks but also dynamically encourages longer, more detailed reasoning ($\mathcal{R}_{\text{extend}}$) when an answer is incorrect. This allows the model to self-correct by scaling its reasoning capacity on-demand, significantly improving reliability and robustness compared to methods with fixed or one-way compression policies.

## 3 METHOD

### 3.1 OVERVIEW

We propose Mixture-of-Length (MoL), a framework that achieves "intelligent brevity" by adaptively modulating response length based on question difficulty. Our method enables models to naturally produce concise responses for simple queries while elaborating reasoning for complex problems. We first introduce a difficulty assessment that quantifies reasoning complexity from cross-document redundancy (proxy-based information-theoretic) (Alon et al., 2003). Based on this assessment, we then propose an adaptive mixture-of-length reasoning approach, driven by a dual-objective reward, to dynamically adjust response length. The overall framework is illustrated in Figure 2.

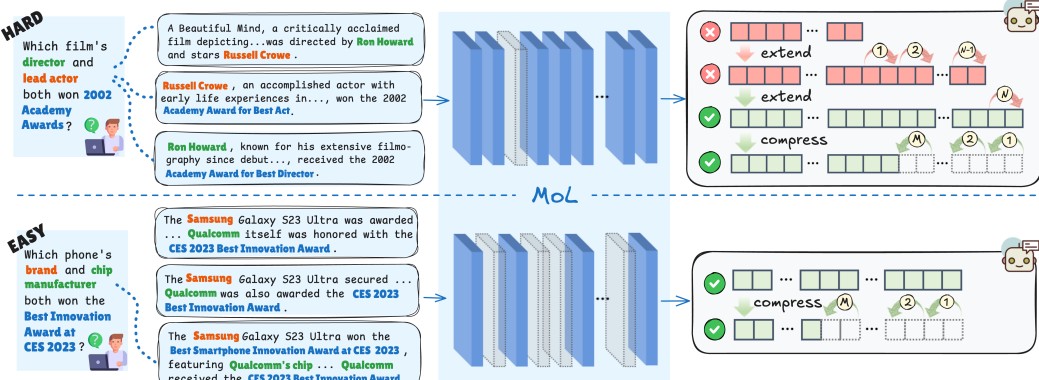

Figure 2: Our framework adaptively controls response length: When a question is answered incorrectly, the model is encouraged by $R_{extend}$ to lengthen its response in order to search for missing evidence chains, whereas once the question is answered correctly, $R_{compress}$ rewards more concise expressions. These two behaviors are adaptively interleaved during training based on the correctness of the current response, rather than forming a fixed multi-stage pipeline.

## 3.2 Difficulty Assessment: From Theory to Implementation

**Conceptual Motivation.** While reasoning complexity is inherently tied to information synthesis across documents, we draw conceptual inspiration from information theory and the classic *Set Cover* problem (Alon et al., 2003; Ash, 2012) to formalize this intuition. To answer a question $q$, a model must synthesize a set of essential knowledge snippets $U = \{u_1, \ldots, u_m\}$, distributed across context documents $D = \{D_1, \ldots, D_n\}$. Each document $D_i$ provides a subset $A_i \subseteq U$ of these snippets.

The complexity of covering all elements in $U$ with a minimal set of documents is closely related to the approximation hardness of the Set Cover problem. However, exact Set Cover computation is NP hard and thus infeasible for large scale QA. We observe that high redundancy, that is, substantial overlap between different $A_i$, corresponds to lower effective complexity, since fewer documents are required to cover $U$, whereas low redundancy corresponds to higher complexity. Motivated by this observation, we propose a practical approximation that characterizes problem difficulty in terms of document redundancy.

**Practical Implementation.** Our heuristic operationalizes redundancy through sentence-level similarity between documents:

**Step 1: Key Information Extraction.** To filter out noise, we first extract question-relevant sentences from each document:

$$D'_i = \{s \in D_i : s \in \text{Top-k}(\text{Sim}(s, q))\}, \tag{1}$$

where Top-k(Sim($s, q$)) denotes the subset of $D_i$ consisting of the k sentences with the highest cosine similarity to q (Sim(s, q) is the cosine similarity between the embeddings of s and q).

**Step 2: Cross-Document Similarity Computation.** We then compute the pairwise similarity between the filtered documents to quantify information redundancy:

$$S_{ij} = \text{cosine}(\text{embed}(D'_i), \text{embed}(D'_j)), \tag{2}$$

The average similarity is calculated as:

$$\bar{S} = \frac{2}{n(n-1)} \sum_{1 \le i < j \le n} S_{ij}, \tag{3}$$

Finally, difficulty is defined as:

$$\mathcal{C}(q, D) = 1 - \bar{S}, \tag{4}$$

**Clarification of Approximation. While our metric is inspired by the complexity formalized in the Set Cover framework through redundancy principles, it is a heuristic approximation tailored for practical QA scenarios. Unlike exact Set Cover solutions, our method avoids combinatorial explosion by focusing on semantic overlap at the sentence level. Empirically, this design achieves strong correlation with difficulty labels documented by humans (81% agreement, Section 4.3), validating its utility as a proxy for reasoning complexity.**

## 3.3 Adaptive Reward Mechanism

Our adaptive reward mechanism can be framed as optimizing the rate-distortion trade-off in reasoning, where we use response length as a proxy for rate and task error as a proxy for distortion.

**Extend Reward for High Distortion.** For complex questions, concise responses often fail to cover the required multi-step reasoning processes. According to Press et al. (2022), many incorrect answers tend to arise when response length is significantly shorter than expected, leading to missing critical reasoning steps. To address this, we design an expansion reward mechanism for incorrect answers (high distortion):

$$\mathcal{R}_{\text{extend}} = \text{clip}\left(\varepsilon_1 - \lambda\left(1 - \frac{L_{\text{actual}}}{L_{\text{target}}}\right), 0, 1\right), \tag{5}$$

where $L_{target}$ denotes the target response length, $L_{actual}$ represents the actual response length, $\varepsilon_1$ is the base reward for incorrect answers, and $\lambda$ controls the reward-length correlation strength. This mechanism encourages longer reasoning to unlock correct paths while incorporating accuracy verification to prevent verbose but ineffective responses.

**Compress Reward for Zero Distortion.** For simple questions, existing models tend to generate responses with redundant explanations and irrelevant information (Sui et al., 2025; Chen et al., 2024), which reduces efficiency and may introduce errors (Zeng et al., 2025). For correct answers (zero distortion), our compression reward encourages finding a minimal-sufficient description, following the Minimum Description Length (MDL) principle (Grünwald, 2007):

$$\mathcal{R}_{\text{compress}} = \text{clip}\left(\varepsilon_2 + \lambda\left(1 - \frac{L_{\text{actual}}}{L_{\text{target}}}\right), 0, 1\right), \tag{6}$$

where $\varepsilon_2$ indicates the base reward for correct answers. This design provides substantial rewards for correct answers while progressively decreasing rewards as response length increases, effectively cultivating the model's ability to "answer on demand" by eliminating non-essential expressions while preserving accuracy.

### 3.4 MIXTURE-OF-LENGTH FRAMEWORK

**Difficulty-Dependent Target Lengths.** Based on the assessed complexity, we assign a target length $L_{\text{target}}$ which acts as an empirical anchor on the rate-distortion curve. The specific length thresholds are initially set based on the response length distribution observed in the HotpotQA dataset(Additional experimental details are provided in Appendix B.5, including additional experiments we conducted demonstrating that MoL can adaptively adjust its behavior, thereby reducing its dependence on the $L_{target}$):

$$L_{\text{target}} = \begin{cases} 512 & \text{if } \mathcal{C}(q, D) \leq 0.3 \text{ (Simple)}, \\ 1024 & \text{if } 0.3 < \mathcal{C}(q, D) < 0.7 \text{ (Medium)}, \\ 2048 & \text{if } \mathcal{C}(q, D) \geq 0.7 \text{ (Complex)}. \end{cases} \tag{7}$$

This anchor is sufficiently long for complex cases, yet short enough to avoid verbosity in simple scenarios. Importantly, we find that different parameter combinations consistently yield performance gains(ablation studies on these parameters are provided in Appendix B.5), indicating that our approach is robust to specific threshold choices.

**Unified Reward Function.** The complete MoL reward implements "intelligent brevity" by dynamically switching between compression and extension modes based on answer correctness. This design prevents reward hacking behaviors where models might exploit the system by generating extremely long or short responses regardless of content quality:

$$R_{\text{MoL}} = \begin{cases} \mathcal{R}_{\text{compress}} & \text{if } y = y^* \text{ (zero distortion)}, \\ \mathcal{R}_{\text{extend}} & \text{if } y \neq y^* \text{ (high distortion)}. \end{cases} \tag{8}$$

**Progressive Learning Strategy.** To ensure stable training, we employ a curriculum learning strategy by dynamically adjusting the length-reward coefficient $\lambda$ over time:

$$\lambda(t) = \max\left(\gamma, \lambda \cdot \left(1 - \frac{t}{T}\right)\right), \tag{9}$$

where $t$ is the current training epoch, $T$ is total epochs, $\lambda$ is a hyperparameter that controls the strength of the correlation between response length and reward, and $\gamma$ is a minimum floor value.

### 3.5 TRAINING OBJECTIVE

We use the GRPO algorithm for optimization. The total reward function combines the standard accuracy reward with our MoL reward:

$$R(x, y) = \alpha \cdot \mathbf{1}[y = y^*] + (1 - \alpha) \cdot R_{\text{MoL}}(x, y), \tag{10}$$

The final optimization objective includes a KL regularizer to a reference policy $\pi_{\text{ref}}$, which stabilizes updates:

$$\mathcal{L}(\theta) = \mathbb{E}_{x, y \sim \pi_\theta}\left[R(x, y) - \beta \log \frac{\pi_\theta(y|x)}{\pi_{ref}(y|x)}\right]. \tag{11}$$

## 4 EXPERIMENTS

**Setup** We utilized a compute node equipped with 64 A100 GPUs for all experiments. Hyperparameters are set as follows: $\alpha = 0.7$ to balance accuracy and efficiency; $\lambda_0 = 1.0, \gamma = 0.3$ for progressive constraint relaxation; $\varepsilon_1 = 0.2, \varepsilon_2 = 0.6$ for the base rewards; and $k = 2$ for sentence selection. We encode the documents with the BGE-M3 encoder. We report experimental results using the F1-score as the primary evaluation metric. All tokens length reported in experimental results refer exclusively to output tokens, excluding input tokens. During training, a prediction is considered correct if its F1-score exceeds 0.8. Additional configurations are detailed in Appendix A. Hyperparameter ablation studies are provided in Appendix B.

**Benchmarks** We performed comprehensive experiments on diverse QA with context tasks, including implicit reasoning, complex reasoning, and long-document reasoning. Our evaluation utilized three benchmark datasets: HotpotQA (Yang et al., 2018), StrategyQA (Geva et al., 2021), and Loong (Wang et al., 2024).

**Baselines** In our experiments, we compare against the following baseline methods: 1) GRPO: A baseline based on standard reinforcement learning. 2) ERPLP: An adaptive resource allocation strategy dynamically adjusts reasoning depth according to problem difficulty (Ling et al., 2025). 3) KIMI: Integrates model distillation, shortest-path sampling, and length penalty to compress long reasoning chains into concise outputs (Team et al., 2025).

### 4.1 COMPARISON EXPERIMENTS

We evaluate the Qwen3-1.7B, Qwen3-8B, Qwen3-14B, and Llama-3.1-8B-Instruct across all benchmark datasets (note that Qwen3-1.7B cannot be evaluated on the Loong dataset due to its context window limitations), with the results presented in Table 1. Experimental results demonstrate that our proposed MoL method achieves intelligent question-adaptive compression through its document-relevance-based dynamic difficulty grading mechanism. For simple questions, it applies substantial compression while preserving more critical information for complex ones, yielding a 49.1% compression ratio with a 6.2% accuracy gain (Qwen3-8B). This adaptive strategy shows significant effectiveness across all three benchmark datasets, not only realizing efficient compression but also substantially improving accuracy. These results suggest that MoL provides a more intelligent and efficient solution for QA with context tasks, effectively addressing a key challenge in traditional methods of balancing response efficiency with answer quality.

The GRPO-trained model indeed achieves improved accuracy, but at the cost of reduced inference efficiency. While ERPLP and KIMI demonstrate certain effectiveness in token compression, their uniform compression strategies exhibit notable limitations. This compression approach particularly impacts the handling of complex questions, excessive compression leads to critical reasoning information loss, resulting in significantly degraded accuracy when answering questions requiring longer reasoning chains. To validate the operational mechanism of the MoL, we illustrate its effect by comparing model outputs before and after training on randomly sampled questions of different difficulty levels. The complete set of examples is available in Appendix F.

### 4.2 ABLATION STUDIES

To validate the effectiveness of our question difficulty assessment method, we conduct comparative experiments on the HotpotQA dataset using three distinct difficulty definition approaches. The experimental results are presented in Table 2. The results demonstrate that models trained with HotpotQA's original difficulty labels achieve superior performance while maintaining output conciseness, confirming the reference value of the dataset's inherent difficulty annotations. In contrast, the passage-based method, which computes difficulty labels through similarity measures between original reference passages, shows suboptimal performance. Specifically, the low-relevance sentences in original reference documents introduce bias to the similarity computation mechanism. The simple questions are then misclassified as difficult ones and consequently leading to unnecessarily verbose model responses. Our proposed method can effectively address such limitations by first extracting question-relevant key sentences before computing similarity. With the improved difficulty assessment, our method has outperformend other methods. Appendix D presents a comparative evaluation of the two methods for computing question difficulty.

Table 1: Performance comparison on QA with Context task under various base LLMs.

| Models | Methods | HotpotQA | | StrategyQA | | Loong | |
|---|---|---|---|---|---|---|---|
| | | Accuracy | Tokens | Accuracy | Tokens | Accuracy | Tokens |
| Qwen3-1.7B | Original | 51.8 | 766 | 81.0 | 397 | - | - |
| | GRPO | 55.5 | 717 | 88.3 | 425 | - | - |
| | ERPLP | 54.4 | 444 | 86.5 | 369 | - | - |
| | KIMI | 55.1 | 475 | 85.9 | 358 | - | - |
| | MoL (Ours) | **57.3** | **384** | **89.3** | **201** | - | - |
| Qwen3-8B | Original | 61.0 | 609 | 93.7 | 468 | 55.8 | 2165 |
| | GRPO | 63.7 | 747 | 95.4 | 478 | 60.8 | 2363 |
| | ERPLP | 63.3 | 394 | 92.0 | 270 | 56.3 | 2037 |
| | KIMI | 62.8 | 444 | 92.5 | 333 | 60.8 | 1938 |
| | MoL(Ours) | **67.2** | **316** | **95.9** | **219** | **62.3** | **1374** |
| Qwen3-14B | Original | 65.7 | 534 | 94.4 | 322 | 62.4 | 1915 |
| | GRPO | 67.2 | 559 | 96.1 | 326 | 71.3 | 1907 |
| | ERPLP | 60.2 | 352 | 94.6 | 248 | 67.0 | 1380 |
| | KIMI | 63.6 | 311 | 92.7 | 304 | 69.6 | 1519 |
| | MoL (Ours) | **69.4** | **298** | **96.8** | **187** | **72.3** | **1128** |
| Llama-3.1-8B-Instruct | Original | 53.3 | 431 | 77.4 | 116 | 36.3 | 742 |
| | GRPO | 64.5 | 1107 | 92.4 | 119 | 57.1 | 896 |
| | ERPLP | 60.2 | 186 | 86.5 | 110 | 49.0 | 158 |
| | KIMI | 64.5 | 196 | 91.6 | 103 | 55.9 | 207 |
| | MoL(Ours) | **69.2** | **169** | **94.1** | **57** | **59.2** | **143** |

Table 2: Performance evaluation of models trained with different difficulty definition strategies on HotpotQA dataset.

| Models | HotpotQA | |
|---|---|---|
| | Accuracy | Tokens |
| Original | 61.1 | 609 |
| Original difficulty | 63.0 | 387 |
| passage | 62.1 | 594 |
| MoL (Ours) | **67.2** | **316** |

Table 3: Comparative analysis of reward mechanisms on model performance (Loong).

| Models | Loong | |
|---|---|---|
| | Accuracy | Tokens |
| Original | 55.8 | 2165 |
| GRPO | 60.8 | 2363 |
| MoL w/o $R_{extend}$ | 58.9 | **1298** |
| MoL w/o $R_{compress}$ | **62.7** | 2862 |
| MoL (Ours) | 62.3 | 1374 |

Our systematic ablation study reveals distinct roles of $R_{compress}$ and $R_{extend}$ (See in Table 3): Removing $R_{extend}$ significantly degrades model accuracy (contrast with GRPO performance), confirming its exclusive contribution to reasoning quality. Conversely, disabling $R_{compress}$ leads to substantially longer outputs with negligible accuracy gains, demonstrating its specialized function in length control. These orthogonal effects collectively validate our reward design's dual-capability architecture: $R_{extend}$ primarily enhances correctness without inducing significant length inflation, while $R_{compress}$ successfully enforces conciseness with minimal sacrifice in accuracy. The ablation studies on the target length and our Progressive Learning Strategy are detailed in Appendix B.5 and Appendix B.6, respectively.

## 4.3 STRATIFIED ANALYSIS BASED ON DIFFICULTY LEVELS

We partition the HotpotQA dataset into ten difficulty-based segments and evaluate both the KIMI method and our approach on each segment, with results shown in Figure 3. Experimental results demonstrate that our method dynamically adjusts token compression strategies according to question difficulty, achieving an optimal balance between performance and efficiency. Our approach yields a significant 7.3% accuracy improvement in high-difficulty segments while maintaining reasonable token counts. For medium-to-low difficulty segments, it achieves superior compression with 10% token reduction while preserving accuracy advantage.

Compared to KIMI's fixed compression strategy, our method exhibits clear difficulty awareness: low-to-medium difficulty questions, we allocate fewer tokens, whereas for high-difficulty questions, increased token allocation mitigates accuracy decline, validating its effectiveness.

To validate the effectiveness of our proposed difficulty assessment method, we employed both the prompt-based model approach and the MoL method to classify the difficulty levels of the HotpotQA

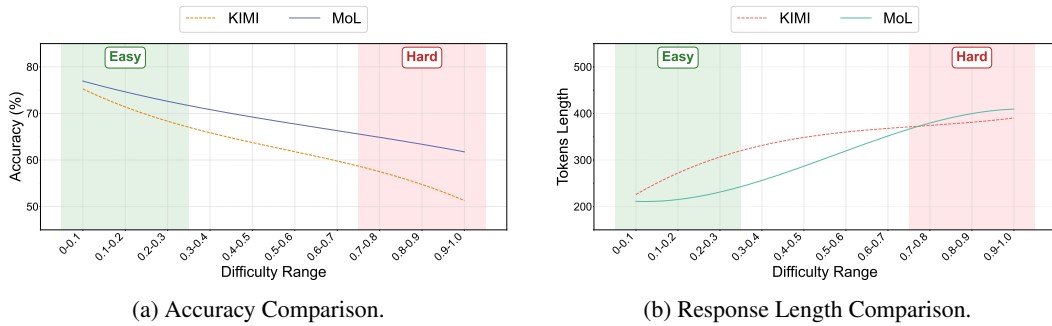

(a) Accuracy Comparison.

(b) Response Length Comparison.

Figure 3: Comparative Analysis of MoL and Kimi Methods by Question Difficulty.

dataset. The experiments utilized DeepSeek-V3 as the classification model, and the performance of the two classification results was evaluated based on the Qwen3-8B model (As shown in Table 4).

Table 4: Comparison of difficulty assessment. MoL demonstrates superior discrimination between easy and hard questions, with a larger accuracy gap (29.1% vs. 8.4%), higher easy-question accuracy, and lower hard-question accuracy, indicating more realistic difficulty evaluation.

| Method | Difficulty (Acc) | | Difference |
| --- | --- | --- | --- |
| | Easy | Hard | |
| Prompt(DS-V3) | 64.1 | 55.7 | 8.4 |
| MoL(Ours) | **79.2(higher)** | **50.1(lower)** | **29.1** |

For datasets classified by difficulty using the Prompt-based method, the model's accuracy did not exhibit significant variation across difficulty levels. In contrast, datasets stratified via the MoL framework exhibited a clear performance gradient reflecting the expected relationship between question difficulty and model accuracy. Specifically, the model performed significantly better on easy questions compared to hard questions, indicating stronger discriminative validity and alignment with real-world difficulty categorization. The MoL methodology successfully identifies simple questions with higher accuracy and challenging questions where accuracy is lower, thereby creating a distinct separation between difficulty tiers.

We further validated the accuracy of the MoL for question difficulty assessment by conducting an expert evaluation on the HotpotQA dataset. The observed agreement rate of 81% between the experts and MoL provides strong evidence for the reliability of our approach. Although difficulty estimation based on cross-document similarity performs well overall, two types of extreme misclassification can occur: samples with high cross-document similarity that nonetheless require multi-step reasoning, and samples with low cross-document similarity whose answers can be directly extracted from a single sentence; in both cases the difficulty estimation method may fail. We include case studies of these two situations in our experiments to demonstrate MoL's self-correction and robustness (see Appendix F.3). We also evaluate the sensitivity of our difficulty estimator to Top-k, embedding model and sentence segmentation; details and results are given in Appendix B.7.

### 4.4 ANALYSIS OF MODEL ACTIVATION PATTERNS

To empirically validate that MoL fosters adaptive computation, we analyzed the model's internal activation patterns. We introduce a relative activation metric to quantify the contribution of each layer (see Appendix C for a detailed definition and implementation). Using this metric, we measured the number of active layers for both simple and difficult problems, with a threshold of $\tau = 0.1$. The results are visualized in Figure 4. The baseline model exhibits a homogeneous activation pattern, engaging a similar number of layers regardless of task difficulty. This confirms its lack of inherent difficulty awareness. In contrast, the MoL-trained model demonstrates significant computational adaptivity: it activates substantially fewer layers for simple problems while recruiting a deeper computational path for difficult ones. This finding provides direct evidence that MoL's training objective leads to an emergent behavior where computational effort is implicitly allocated based on perceived problem complexity, which is a fundamental driver of its efficiency and performance gains.

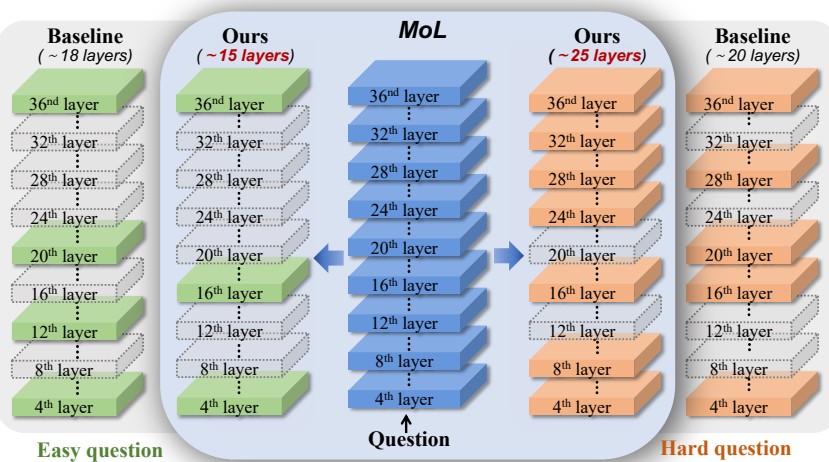

Figure 4: Baseline vs. MoL layer activation on easy vs. hard questions: colored bars denote activated layers; uncolored bars denote inactive layers (Analysis on the HotpotQA dataset).

To demonstrate the generalization capability of our approach, we evaluate models trained on the HotpotQA dataset across five benchmark datasets. As shown in Table 5, the Qwen3-8B model trained with MoL exhibits remarkable generalization performance: on unseen StrategyQA and Loong datasets, it not only maintains high accuracy rates of 94.4% and 57.2% respectively, but also compresses the average response length to 239 and 1574 tokens, achieving reductions of 48.9% and 27.3% compared to the original model. These results strongly validate that MoL enables models to autonomously recognize question difficulty and adaptively select response strategies. Crucially, the stark contrast with the GRPO baseline confirms that the performance gains are attributable to the MoL framework itself, rather than to reinforcement learning in general.

## 4.5 GENERALIZATION STUDIES

Table 5: Qwen3-8B performance across five datasets when trained on HotpotQA.

| Models | HotpotQA | | StrategyQA | | Loong | | CQA | | SVAMP | |
|---|---|---|---|---|---|---|---|---|---|---|
| | Acc | Tokens | Acc | Tokens | Acc | Tokens | Acc | Tokens | Acc | Tokens |
| Original | 61.0 | 609 | 93.7 | 468 | 55.8 | 2165 | 66.5 | 673 | 93.7 | 1397 |
| GRPO | 63.7 | 747 | 92.9 | 513 | 56.4 | 2276 | 67.1 | 647 | 93.1 | 1436 |
| MoL | **67.2** | **316** | **94.4** | **239** | **57.2** | **1574** | **68.2** | **542** | **94.6** | **1129** |

Our model's training paradigm is based on document-grounded datasets, which necessitates an investigation into its applicability to standard, non-document-grounded tasks. To this end, we performed evaluations on two benchmarks: a commonsense reasoning dataset (CommonsenseQA) (Talmor et al., 2019) and a mathematical word problem dataset (SVAMP) (Patel et al., 2021). The empirical results are summarized in Table 5. Notably, our method exhibits strong performance, indicating that the MoL training framework endows the model with an inherent capability to gauge question difficulty and dynamically allocate computational budget for generating responses. This outcome strongly validates the generalization power of our approach beyond its training domain.

## 4.6 LONG CONTEXT SCENARIO

We evaluate model performance on long contexts by testing on data divided into two length intervals: 10k-50k and 50k-100k. As demonstrated in Figure 5. Results show our method achieves the best overall performance in both intervals, particularly excelling at complex reasoning tasks. This superior performance stems from MoL's adaptive $L_{target}$ parameter selection, which enables optimal response strategies for texts of varying lengths. Notably, our method maintains comparable

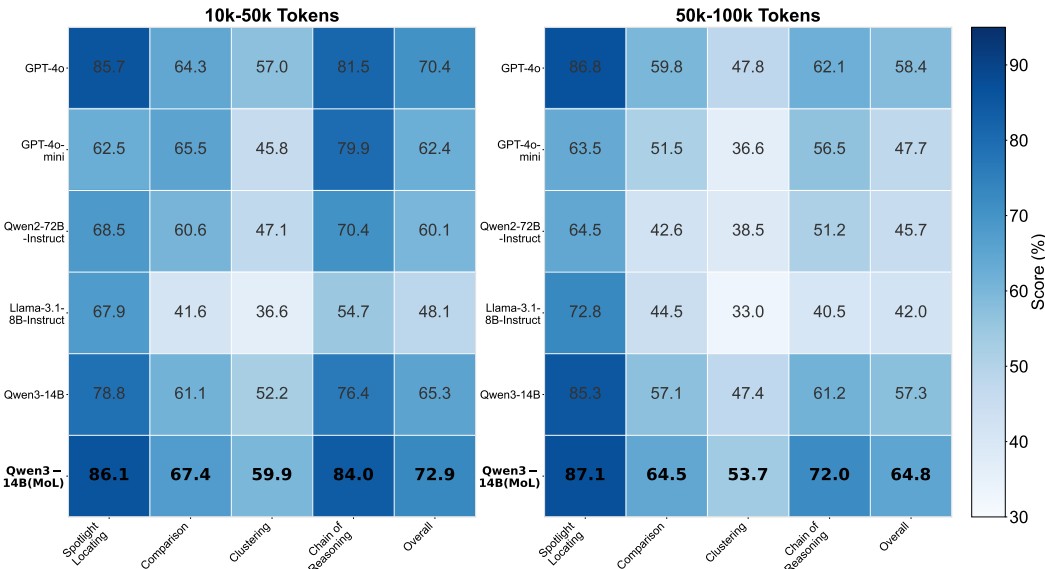

Figure 5: Model performance comparison on the Loong dataset (10-50k and 50-100k).

performance between the 50k-100k and 10k-50k ranges, demonstrating significantly better stability than other baseline models. Experimental results demonstrate that our difficulty-adaptive approach achieves precise difficulty assessment through sentence decomposition while maintaining robustness to text length. Our case analysis shows that, even when desired lengths are shorter than initially specified, the method generates suitable responses, evidencing the effectiveness of the length-constraint relaxation mechanism and robustness to varying length requirements.

## 5 CONCLUSION

We propose Mixture-of-Length (MoL), a novel framework that dynamically adapts response lengths to problem difficulty, effectively balancing reasoning depth and efficiency in QA with context tasks. MoL combines a principled difficulty assessment with a dual-objective reward mechanism, which gives rise to an emergent "intelligent brevity" behavior. Our post-hoc analysis shows this adaptability correlates with the model's internal layer contributions, emulating human-like cognitive efficiency. MoL represents a promising paradigm for scalable, context-aware reasoning in LLMs and suggests promising directions for developing intelligent, resource-efficient QA systems.

## 6 ETHICS STATEMENT

This study uses only publicly available datasets that contain no personal or sensitive identifying information. We conducted manual difficulty annotations for a subset of samples; annotators were informed of the research purpose and voluntarily consented to participate. A comprehensive description of our use of LLMs is documented in Appendix E.

## 7 LIMITATIONS

**Analysis is Correlational:** The observed link between shorter outputs and fewer activated layers is post-hoc and does not establish causality. More rigorous mechanistic studies are needed to confirm how MoL influences internal computation.

**Generalization Constraints in Multi Document Dependent Difficulty Assessment:** We acknowledge that the proposed difficulty assessment method is inherently tied to multi document tasks which limits its direct applicability to single document or document free scenarios. While the bidirectional reward function in MoL demonstrates task agnostic properties (e.g. outperforming baselines on

CommonsenseQA and SVAMP in Table 5) its adaptation to non multi document settings currently relies on heuristic initial target lengths derived from approximate output token ranges. Future work will focus on developing a unified difficulty assessment framework applicable across diverse task modalities including single document QA and document agnostic reasoning.

## 8 REPRODUCIBILITY STATEMENT

We have taken several steps to facilitate independent verification of our results. The main paper details the model architecture and training objectives (see Section 3), Experimental Setup and datasets (Section 4), and ablation configurations (Appendix B). Complete hyperparameter lists, hardware/software specifications are documented in Appendix A. Dataset processing pipelines are described in Appendix G. The source code implementing our method will be submitted as part of our supplementary materials.

## 9 ACKNOWLEDGMENTS

This research was partially supported by National Natural Science Foundation of China under grants No. T2541004, Zhejiang Key R&D Program of China under grant No. 2025C02120, No. 2024SSYS0026, Zhejiang Key Laboratory of Medical Imaging Artificial Intelligence, and the Transvascular Implantation Devices Research Institute (TIDRI) under Grant No. KY052025003 and the Ant Group Research Intern Program.

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

## A  EXPERIMENTS

**Setup**  Our experiments were conducted using 64 A100 with 80GB memory per device. Hyperparameters are set as follows: balancing coefficient $\alpha = 0.7$, follow other work. length-correlation coefficient $\lambda = 1$, $\gamma = 0.3$, base rewards $\varepsilon_1 = 0.2$ and $\varepsilon_2 = 0.6$. To ensure accurate retrieval of the most relevant documents, we set the hyperparameter $k = 2$ in the $Top - k$ algorithm during the document pruning stage, selecting the top two highest-scoring sentences per document based on their relevance to the query. We propose a dynamic length selection mechanism with target length $L_{target}$ adapting to question difficulty levels: for simple questions (typically factual queries), we set $L_{target} = 512$ to capture core information needs; for medium-difficulty questions (often involving multi-clause analysis), $L_{target} = 1024$ to incorporate contextual dependencies; and for complex questions (requiring cross-document reasoning), $L_{target} = 2048$ to enable comprehensive modeling. This tiered design ensures computational efficiency while optimizing semantic modeling capacity across complexity levels. Here are some specific training configurations:

Table 6: Hyperparameters for experiments.

| Configuration | Value |
|---|---|
| Number of epochs | 3 |
| Devices | $64 \times$ A100 |
| Total Batch size | 256 |
| Learning rate | $5 \times 10^{-5}$ |

Each experimental run required approximately ten hours to complete. Importantly, the similarity computation occurs offline during training data preparation and does not introduce any computational overhead during model inference.

**Benchmarks** We performed comprehensive experiments on diverse QA with context tasks, including implicit reasoning, complex reasoning, and long-document reasoning. Our evaluation utilized three benchmark datasets: HotpotQA, StrategyQA, and Loong.

**HotpotQa:** A benchmark designed to advance research in complex reasoning and interpretable question answering systems. It contains Wikipedia-based questions and answers where each problem requires integrating information from multiple documents to derive solutions, with sentence-level supporting facts provided as supervisory signals. The dataset features diverse question types including fact comparison questions and serves as an effective benchmark for evaluating models' multi-hop reasoning capabilities.

**StrategyQa:** A Boolean question answering dataset focusing on implicit reasoning, containing 2,780 yes/no questions requiring multi-step inference to resolve. Distinct from explicit multi-hop QA tasks, this dataset challenges models to autonomously infer problem-solving strategies (e.g., temporal comparisons, logical deductions) without exposing intermediate reasoning steps in the questions. Each instance is annotated with decomposed reasoning steps and corresponding Wikipedia evidence paragraphs, which can guide model learning of complex inference processes through supervised training.

**Loong:** An innovative long-text multi-document question answering benchmark designed to evaluate large language models (LLMs) in real-world long-context comprehension scenarios. This dataset ensures models must comprehensively understand all documents by distributing answer-related evidence across multiple passages. It encompasses three domains: financial reports, legal cases, and academic papers, covering four task types (focused localization, comparison, clustering, and reasoning chains), and provides test sets with varying lengths ranging from 10K to 250K tokens.

## B  HYPERPARAMETER ANALYSIS

To systematically investigate the impact of key hyperparameters, we conduct an ablation study on the HotpotQA dataset using the Qwen3-8B model.

### B.1  ANALYSIS OF THE TRADE-OFF COEFFICIENT $\alpha$ IN THE REWARD FUNCTION

The balancing coefficient $\alpha$ is a key hyperparameter in the reward function that modulates the trade-off between model accuracy and efficiency. The results, presented in Table 7, delineate how the variation in $\alpha$'s value affects the final performance.

**Low $\alpha$ values:** The reward is dominated by the length term ($R_{MoL}$), prioritizing extreme conciseness. However, this leads to the omission of critical reasoning steps, resulting in suboptimal accuracy due to excessive compression.

**High $\alpha$ value:** The accuracy term ($R_{accurate}$) dominates, prompting the model to adopt a conservative strategy to maximize correctness. This generates verbose reasoning chains with redundant steps, causing a significant increase in token consumption while the accuracy plateaus, yielding no further gains.

Table 7: Experimental Results with Different Values of Parameter $\alpha$.

| $\alpha$ | Accuracy | Tokens |
|---|---|---|
| 0.1 | 64.9 | **264** |
| 0.3 | 66.3 | 301 |
| 0.5 | 67.0 | 313 |
| 0.7 | 67.2 | 316 |
| 0.9 | **67.3** | 371 |

**Optimal value:** The model learns to first secure high reward by ensuring correctness and then optimize for conciseness. We ultimately select $\alpha$=0.7 as the default configuration, as it achieves high accuracy while maintaining near-optimal efficiency.

These findings validate the effectiveness and controllability of our proposed reward mechanism. The coefficient $\alpha$ serves as a reliable and intuitive "tuning knob" for the accuracy-efficiency trade-off. Furthermore, the presence of a clear optimum demonstrates that our dual-objective reward design is necessary for successfully balancing these two competing goals.

## B.2 ANALYZING THE EFFECT OF CONSTRAINT STRENGTH $\gamma$

The hyperparameter $\gamma$ controls the minimum strength of the length constraint enforced during the late training phase. As shown in Table 8, our ablation study demonstrates the impact of $\gamma$ on the final performance.

Table 8: Experimental Results with Different Values of Parameter $\gamma$.

| $\gamma$ | Accuracy | Tokens |
|---|---|---|
| 0.1 | 66.9 | 352 |
| 0.3 | 67.2 | **316** |
| 0.5 | **67.4** | 372 |
| 0.7 | 66.7 | 391 |
| 0.9 | 65.1 | 536 |

**Low $\gamma$ values:** The constraint strength decays too rapidly in the late training phase, causing the model to lose almost all motivation to optimize response length and regress to its inherent verbose generation pattern. Consequently, token usage is not minimized, and the accuracy remains suboptimal as the model fails to adequately learn the reward signal associated with conciseness.

**Optimal $\gamma$ values:** The model remains under a moderate length constraint in the late stage, successfully internalizing the strategy of "pursuing conciseness while guaranteeing correctness." This leads to a superior balance between accuracy and efficiency.

**High $\gamma$ values:** An excessively strong constraint throughout the entire training process inhibits the model's exploration and generalization capability, particularly for complex questions requiring longer reasoning chains. The model struggles to generate necessary intermediate steps, resulting in a significant drop in accuracy. Meanwhile, to meet the stringent length limit, the model may produce obscure, abnormally high-density text, which can paradoxically lead to an increased token count.

Conclusion: $\gamma$ acts not as a simple intensity parameter, but as a critical regulator between stability and flexibility. Our experiments demonstrate that a moderate $\gamma$ value is essential: it prevents the constraint from vanishing too early to ensure stable convergence, while also avoiding overly strong restrictions to preserve the flexibility needed for solving complex problems, ultimately leading to a synergistic improvement in both accuracy and efficiency.

## B.3 THE NON-MONOTONIC IMPACT OF $\varepsilon_1$ AND PARETO-OPTIMAL CHOICE

$\varepsilon_1$ sets the base reward value granted for producing extended responses. As shown in Table 22, our ablation study demonstrates the impact of $\varepsilon_1$ on the model's performance.

Table 9: Experimental Results with Different Values of Parameter $\varepsilon_1$.

| $\varepsilon_1$ | Accuracy | Tokens |
|---|---|---|
| 0 | **67.4** | 362 |
| 0.2 | 67.2 | 316 |
| 0.4 | 66.7 | **311** |
| 0.6 | 65.6 | 379 |
| 0.8 | 65.1 | 413 |

The results reveal a non-monotonic trend, highlighting the delicate trade-off in the design of $\varepsilon_1$:

$\varepsilon_1 = 0$: This configuration achieves the highest accuracy . Since the model receives no base reward for generating long but incorrect answers, the expansion reward is solely determined by the alignment with the target length (i.e., "precise expansion"). This forces the model to be highly efficient and precise in its remedial reasoning, filtering out more effective reasoning paths and indirectly boosting final accuracy. However, this strong constraint also slightly limits the model's expressive capacity, preventing it from achieving the highest efficiency.

$\varepsilon_1 \in [0.2, 0.4]$ (Low Range): In this range, the model incurs only a minimal and acceptable drop in accuracy while achieving substantial gains in efficiency. A modest $\varepsilon_1$ in this interval provides a "safety net," encouraging beneficial exploration when uncertain without incentivizing meaningless verbosity. It works synergistically with the compression reward to steer the model toward the global optimum of being both correct and concise.

$\varepsilon_1 \geq 0.6$ (High Range): A significant performance degradation is observed, with both accuracy and efficiency declining. An excessively high $\varepsilon_1$ distorts the reward signal, teaching the model a harmful shortcut: generating a long incorrect answer yields a higher reward than a short incorrect one. This effectively incentivizes the model to produce "knowingly wrong" verbose outputs instead of striving for correctness, leading to overall performance deterioration.

Conclusion and Choice: Our experiment indicates that tuning $\varepsilon_1$ requires a balance between providing exploratory freedom and avoiding reward distortion. Although $\varepsilon_1$=0 yields the peak accuracy, we ultimately select $\varepsilon_1$=0.2 as the default. The rationale is that this value achieves the best Pareto frontier for overall performance, trading a negligible accuracy drop for the largest efficiency gain. This finding indicates that a small but non-zero $\varepsilon_1$ is crucial for an effective error-tolerance mechanism.

## B.4 ABLATION STUDY ON THE COMPRESSION REWARD $\varepsilon_2$

$\varepsilon_2$ sets the base reward value granted for producing compressed responses. As shown in Table 10, our ablation study demonstrates the impact of $\varepsilon_2$ on the model's performance.

Table 10: Experimental Results with Different Values of Parameter $\varepsilon_2$.

| $\varepsilon_2$ | Accuracy | Tokens |
|---|---|---|
| 0 | 61.7 | 186 |
| 0.2 | 64.3 | 264 |
| 0.4 | 65.9 | 292 |
| 0.6 | 67.2 | 316 |
| 0.8 | 67.7 | 441 |

At $\varepsilon_2 = 0$: The model exhibits harmful over-compression due to the absence of a base reward for correctness, causing it to over-optimize for the length bias term. This results in an extremely low

token count but a significant degradation in accuracy, indicating the omission of critical reasoning steps.

For $\varepsilon_2 \in [0.2, 0.6]$: The model receives a strong positive signal that "correctness yields high reward." This drives a learning strategy that prioritizes answer accuracy before optimizing for conciseness, leading to a Pareto improvement in both metrics. Accuracy is substantially improved with only a moderate increase in token usage.

At $\varepsilon_2 = 0.8$: An excessively high base reward distorts the optimization objective. The model adopts a conservative strategy, generating verbose chains of reasoning to ensure correctness. This results in a sharp increase in token count and a complete loss of efficiency gains.

Conclusion and Selection: These findings demonstrate that $\varepsilon_2$ is a pivotal hyperparameter for balancing accuracy and conciseness. Based on a comprehensive evaluation, we select $\varepsilon_2 = 0.6$ as the optimal configuration. This value achieves near-peak accuracy while maintaining responses within an efficient length range, thereby achieving the best overall performance balance.

## B.5 ABLATION STUDY ON THE TARGET LENGTHS $L_{target}$

To establish optimal target length thresholds, we conducted a systematic analysis on the first 1,000 samples from the HotpotQA dataset. First, We use the Qwen3-8B model to directly classify the difficulty of questions. Subsequently, we analyzed the distribution of output lengths generated by the model for each difficulty tier, with the results visualized in Figure 6. Through statistical analysis of these distributions, we empirically determined the target lengths as $L_{target} = \{512, 1024, 2048\}$, corresponding to simple, medium, and complex questions respectively. This data-driven approach ensures that the thresholds align with the inherent reasoning complexity while maintaining computational efficiency.

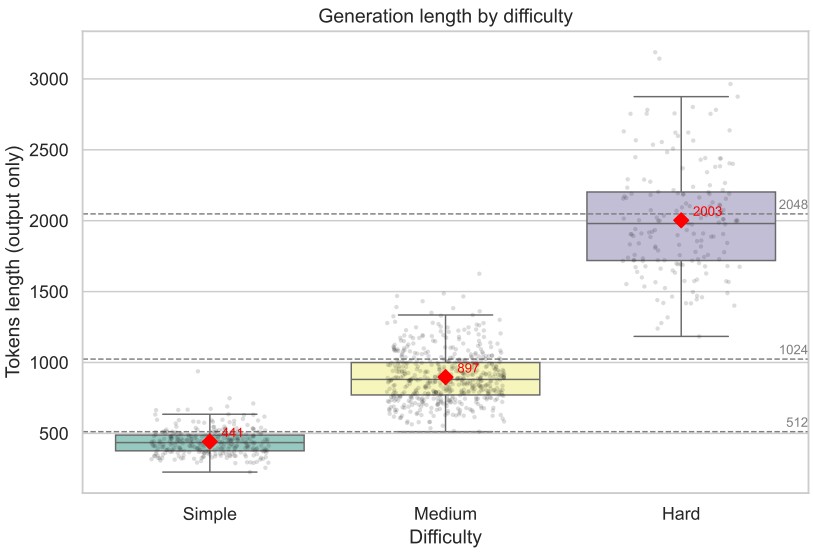

Figure 6: Distribution of generated output lengths by difficulty class (Simple / Medium / Hard). Tokens count only model outputs (exclude inputs), red diamonds mark class means.

To assess how the initial target length settings for problems of varying difficulty affect performance, we conducted an additional experiment using four alternative sets of $L_{target}$ values with the Qwen3-8B model:

$$L_{\text{target}} = \begin{cases} 256 & \text{Simple,} \\ 512 & \text{Medium,} \\ 1024 & \text{Complex,} \end{cases} \qquad L_{\text{target}} = \begin{cases} 256 & \text{Simple,} \\ 256 & \text{Medium,} \\ 256 & \text{Complex,} \end{cases}$$

$$L_{\text{target}} = \begin{cases} 512 & \text{Simple,} \\ 512 & \text{Medium,} \\ 512 & \text{Complex,} \end{cases} \qquad L_{\text{target}} = \begin{cases} 1024 & \text{Simple,} \\ 1024 & \text{Medium,} \\ 1024 & \text{Complex,} \end{cases} \qquad (12)$$

The results, summarized in Table 11, show that the choice of initial target lengths has a negligible impact on final outcomes. These findings validate the effectiveness of our Progressive Learning Strategy: the target response length functions primarily as a guiding signal in the early stages of training to shape response lengths, but it does not materially influence the final answers. Consequently, our approach remains robust across a broad spectrum of problem difficulties, demonstrating strong generalization.

Table 11: Experimental Results with Different Values of Parameter $L_{\text{target}}$.

| Methods | HotpotQA | | StrategyQA | | Loong | |
|---|---|---|---|---|---|---|
| | Acc | Tokens | Acc | Tokens | Acc | Tokens |
| Original | 61.0 | 609 | 93.7 | 468 | 55.8 | 2165 |
| MoL(256,512,1024) | 66.7 | 303 | 96.1 | 247 | 61.4 | 1473 |
| MoL(512,1024,2048) | 67.2 | 316 | 95.9 | 219 | **62.3** | 1374 |
| MoL(256,256,256) | 67.0 | 303 | **96.4** | 236 | 61.7 | **1338** |
| MoL(512,512,512) | **67.4** | 331 | 95.9 | **213** | 62.1 | 1402 |
| MoL(1024,1024,1024) | 66.9 | **286** | 96.2 | 225 | 61.3 | 1425 |

To investigate the impact of $L_{target}$ on model training efficiency, we present a systematic evaluation of the Qwen3-8B model's training dynamics on the HotpotQA dataset under varying initial target length configurations, focusing on performance variations across epochs. Figure 7 and Figure 8 illustrate the evolution of Accuracy and generated token length during training, respectively. Experimental results demonstrate that larger deviations between the initial target length and the optimal value necessitate more training steps to reach convergence. However, as training progresses, the influence of target length diminishes, with all configurations eventually converging to comparable performance levels. This indicates that while the initial length setting impacts training efficiency, it has minimal effect on the model's final generalization capability.

### B.6 ABLATION STUDY ON THE PROGRESSIVE LEARNING STRATEGY

To validate the effectiveness of our Progressive Learning Strategy, we conducted an ablation study by removing this module and re-evaluating the model's performance. The results are presented in Table 12.

Table 12: Experimental Results with Progressive Learning Strategy.

| Methods | HotpotQA | | StrategyQA | | Loong | |
|---|---|---|---|---|---|---|
| | Acc | Tokens | Acc | Tokens | Acc | Tokens |
| Original | 61.0 | 609 | 93.7 | 468 | 55.8 | 2165 |
| MoL w/o $\lambda(t)$ | 65.3 | 835 | 93.1 | 592 | 49.1 | **913** |
| MoL | **67.2** | **316** | **95.9** | **219** | **62.3** | 1374 |

The results reveal that the Progressive Learning Strategy has a substantial impact on performance. Without this strategy, the model's response length is severely constrained by the pre-defined initial target value, failing to dynamically adapt to the problem's complexity. This limitation is particularly detrimental to complex problems requiring long reasoning chains, such as those in the Loong dataset,

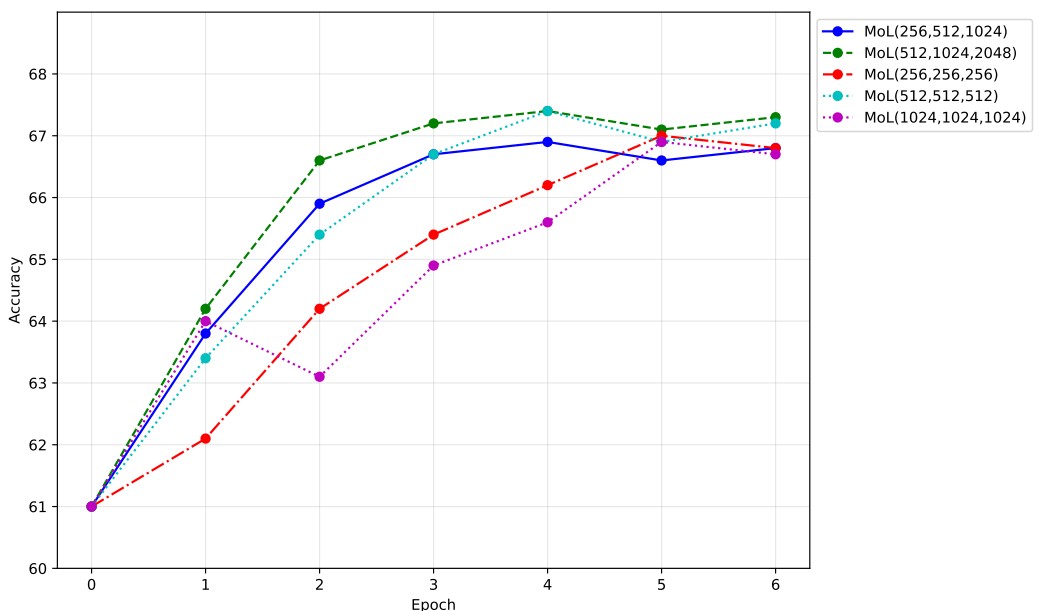

Figure 7: Evolution of accuracy across training epochs.

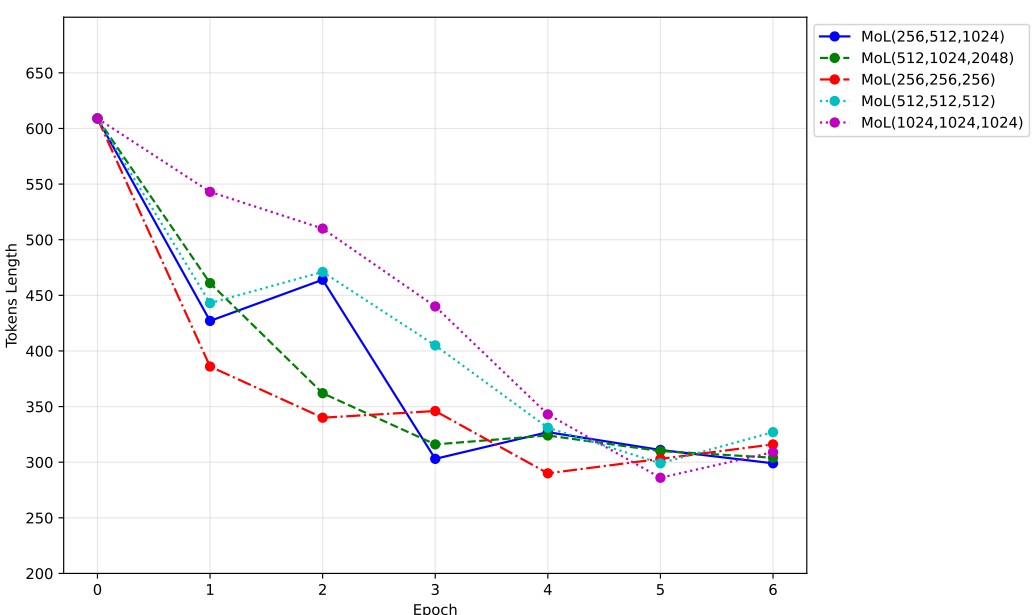

Figure 8: Evolution of model output length across training epochs.

leading to a drastic performance drop. Furthermore, this inability to adapt the output length also impairs performance on simpler tasks, resulting in decreased accuracy across all datasets. These findings firmly demonstrate that the Progressive Learning Strategy is an indispensable component for generating high-quality and adaptive responses.

### B.7 ABLATION STUDY ON THE ROBUSTNESS OF DIFFICULTY ESTIMATION

To evaluate the sensitivity of our difficulty estimator to Top-k retrieval, the embedding encoder, and sentence segmentation, we ran controlled ablations on the Loong dataset using Qwen3-8B. Ta-

ble 13 summarizes the results for varying Top-k (k=1,2,3), Table 14 compares two sentence encoders (Sentence-T5 and BGE-M3), and Table 15 reports results for two segmentation granularities (1 sentence vs. 2 sentences per retrieval unit). Each entry reports downstream answer accuracy and the average number of generated tokens (output only). In brief, k=2 yields a good trade-off between accuracy and generation length (k=1 produces shorter outputs but slightly lower accuracy), encoder choice has only a minor effect on accuracy, and merging two sentences modestly increases generated length without materially changing accuracy. Overall, MoL is robust to these design choices: they have minimal impact on the final results.

Table 13: Top-k ablation on Loong (Qwen3-8B). Impact of varying Top-k retrieval (k=1,2,3) on downstream answer accuracy and average generated tokens (output only). "Original" denotes the baseline without MoL.

| Methods | Loong | |
|---|---|---|
| | Acc | Tokens |
| Original | 55.8 | 2165 |
| MoL(k=1) | 61.7 | **1199** |
| MoL(k=2) | **62.3** | 1374 |
| MoL(k=3) | 62.1 | 1422 |

Table 14: Embedding encoder ablation on Loong (Qwen3-8B). Comparison between Sentence-T5 and BGE-M3 sentence encoders reporting answer accuracy and average output token length.

| Methods | Loong | |
|---|---|---|
| | Acc | Tokens |
| MoL(Sentence-T5) | **62.6** | 1462 |
| MoL(BGE-M3) | 62.3 | **1374** |

Table 15: Sentence granularity ablation on Loong (Qwen3-8B). Effect of segmentation unit (single-sentence vs. two-sentence retrieval units) on downstream accuracy and average generated tokens.

| Methods | Loong | |
|---|---|---|
| | Acc | Tokens |
| Original | 55.8 | 2165 |
| MoL(one sentence) | 62.3 | **1374** |
| MoL(two sentence) | **62.5** | 1439 |

## C  IMPLEMENTATION DETAILS FOR LAYER-WISE ACTIVATION

### C.1  QUANTIFYING LAYER-WISE ACTIVATION

To investigate the computational dynamics of our MoL-trained model during inference, we quantify each Transformer's layer-wise activation via its relative contribution to the residual stream.

**Motivation.** Relative activation is preferable to absolute magnitudes because it: (i) normalizes for scale differences across layers induced by residual connections, (ii) captures the proportional change each layer makes to the information flow, and (iii) remains comparable across model sizes and architectures.

**Definition.** Consider a Pre-LN decoder-only Transformer layer $l$ operating on an input residual stream $\mathbf{r}_{\text{in}}^{(l)} \in \mathbb{R}^{T \times D}$. Let $\boldsymbol{\Delta}_{\text{attn}}^{(l)}$ and $\boldsymbol{\Delta}_{\text{mlp}}^{(l)}$ denote, respectively, the actual updates added back to the residual stream by the attention and MLP submodules. The total update is $\boldsymbol{\Delta}_{\text{layer}}^{(l)} = \boldsymbol{\Delta}_{\text{attn}}^{(l)} + \boldsymbol{\Delta}_{\text{mlp}}^{(l)}$. We define the relative activation as:

$$\alpha^{(l)} = \frac{\text{RMS}(\boldsymbol{\Delta}_{\text{layer}}^{(l)})}{\text{RMS}(\mathbf{r}_{\text{in}}^{(l)}) + \epsilon} \tag{13}$$

where $\epsilon$ is a small constant for numerical stability. A layer is considered active if $\alpha^{(l)} > \tau$ (threshold $\tau$ described in Section C.3).

## C.2 Implementation Details

**Masked RMS for padded batches.** For a tensor $\mathbf{x} \in \mathbb{R}^{B \times T \times D}$ and an attention mask $\mathbf{m} \in \{0, 1\}^{B \times T}$, we compute a masked Root Mean Square (RMS) over all valid elements:

$$\text{masked\_rms}(\mathbf{x}, \mathbf{m}) = \sqrt{\frac{\sum(\mathbf{x}^2 \odot \mathbf{m}')}{(\sum \mathbf{m}) \cdot D} + \epsilon} \tag{14}$$

where $\mathbf{m}'$ is the mask $\mathbf{m}$ broadcast to the shape of $\mathbf{x}$ (i.e., $[B, T, 1]$), and $\odot$ denotes element-wise multiplication. We compute in float32 and clamp the denominator to avoid division-by-zero when all tokens are padding.

**Non-invasive residual differencing.** To ensure that the measured updates match the *actual* tensors added to the residual stream (including any internal dropout, scaling, or gating), we avoid reading from intermediate projection layers and instead compute updates by differencing the residual states:

$\mathbf{r}_0^{(l)}$:block input before LayerNorm (block-level forward pre-hook),

$\mathbf{r}_1^{(l)}$:after attention residual addition and before MLP (captured as the input to the post-attention LayerNorm via its forward pre-hook),

$\mathbf{r}_2^{(l)}$:block output (block-level forward hook).

Then

$$\boldsymbol{\Delta}_{\text{attn}}^{(l)} = \mathbf{r}_1^{(l)} - \mathbf{r}_0^{(l)}, \tag{15}$$

$$\boldsymbol{\Delta}_{\text{mlp}}^{(l)} = \mathbf{r}_2^{(l)} - \mathbf{r}_1^{(l)}, \tag{16}$$

$$\boldsymbol{\Delta}_{\text{layer}}^{(l)} = \mathbf{r}_2^{(l)} - \mathbf{r}_0^{(l)}. \tag{17}$$

This construction is architecture-robust and aligns the numerator and denominator of Eq. 13 to the same residual stream.

**Measurement protocol.** We instrument models in evaluation mode under no-gradient execution to disable dropout and reduce overhead. For mixed precision, we upcast to float32 for statistics. Unless otherwise stated, we use $\epsilon = 10^{-8}$ for FP32 and $10^{-6}$ for FP16/BF16.

## C.3 Threshold Selection and Robustness

**Threshold.** We set $\tau = 0.1$ based on empirical analysis across multiple model sizes and reasoning tasks. This value provides effective separation between layers with meaningful contributions and those with minimal updates.

**Cross-architecture verification.** We verified LLaMA and Qwen style Pre-LN decoders. For other variants (e.g., Post-LN), the same principle applies: capture the residual states immediately before and after each residual addition to form $\boldsymbol{\Delta}_{\text{attn}}$ and $\boldsymbol{\Delta}_{\text{mlp}}$ via differencing (Eq. 15). This guarantees inclusion of any internal dropout, scaling, or gating before residual addition.

## D  PARAGRAPH-LEVEL VS. SENTENCE-LEVEL SIMILARITY MATCHING

Through controlled experiments comparing paragraph-level and sentence-level similarity matching (Results in Tables 16 and Table 17), we identify fundamental limitations in paragraph-level approaches: they inherently incorporate numerous low-relevance sentences (particularly question-related expository content). These noisy segments systematically distort similarity computation, causing significant underestimation of semantic alignment and consequent overestimation of question difficulty. Our proposed Top-k key sentence filtering prior to similarity calculation demonstrably mitigates this issue, with experimental results validating the efficacy of our approach.

## E  THE USE OF LARGE LANGUAGE MODELS (LLMS)

All models utilized in this work are publicly available. We employed them solely for language polishing to improve the readability of our text. It is important to note that these models were not involved in any scientific decision-making. Furthermore, all model-assisted outputs underwent rigorous human review to ensure compliance with ethical and legal standards.

## F  CASE STUDY

### F.1  ANALYSIS OF RESPONSE PATTERNS FOR SIMPLE QUESTIONS

When handling relatively simple questions that require no reasoning (e.g., asking whether visitors are allowed to use mobile phones to take photos or videos during the ride in the "Jurassic World Adventure" attraction at Universal Beijing Resort), baseline models tend to generate overly verbose responses (Results in Tables 18). Although they accurately output the core information that electronic devices are prohibited for photography, the models still append redundant content, such as repeatedly mentioning instructions from official channels. Textual analysis reveals that such expansions primarily stem from verbatim extraction of reference documents; while semantically correct, they fail to meet the task requirement of concise responses. In contrast, models trained with MoL demonstrate precise response control, strictly confining their outputs to the core information sought by the question, ensuring answer accuracy while significantly improving response efficiency.

### F.2  ANALYSIS OF REASONING CAPABILITIES FOR COMPLEX QUESTIONS

The baseline models exhibit fundamental limitations in processing multi-conditional reasoning tasks, as evidenced by their performance on greenhouse gas effect questions (Results in Tables 19). While correctly identifying the basic absorption ratios of $CH_4$ to $CO_2$ (84×) and $NF_3$ to $CO_2$ (16,100×), these models fail to incorporate the critical constraint regarding fluorinated gases' 272× radiative efficiency relative to $CH_4$, resulting in erroneous linear extrapolations. In contrast, our MoL-enhanced model demonstrates superior information integration and quantitative reasoning capabilities. It successfully captures all relevant document constraints, establishes cross-sentence numerical relationships, and performs the necessary multi-step calculation (16,100/84×272) to accurately determine $NF_3$'s superior absorption efficacy. This performance improvement confirms that our approach not only enhances key information extraction completeness but also develops the model's capacity for evidence-based quantitative reasoning, representing a significant advancement in complex scientific question answering.

### F.3  ROBUSTNESS TO DIFFICULTY MISCLASSIFICATION (MoL SELF-CORRECTION)

Although difficulty estimation based on cross-document similarity works in most cases, extreme misclassification can still occur. Table 20 and Table 21 present two types of extreme misclassification scenarios and MoL's self-correction behavior. Table 20 illustrates the case of high cross-document similarity but actually requiring multi-step reasoning: because the mean similarity $\bar{S}$ is large, the example is initially judged as simple, so the model attempts a short answer and makes an error (the table shows the initial output). When the answer is judged incorrect, MoL's extension reward $R_{extend}$ encourages the model to produce longer, more complete chains of reasoning

Table 16: A Case Study on Estimating Question Difficulty Through Sentence-Level Similarity Analysis.

| Sentence-level Similarity Matching | |
|---|---|
| Query | Who is the author of The Story of the Stone? |
| Doc | Doc1: "The Story of the Stone, recognized as the foremost of China's Four Great Classical Novels, was authored by the Qing dynasty writer Cao Xueqin... Beyond its narrative depth, the work provides a critical depiction of feudal society's moral decay and social intricacies. With its vast ensemble of characters and richly woven plotlines, it has been acclaimed as the "Encyclopedia of Chinese Feudal Society." Doc2: "Cao Xueqin, the author of the Story of the Stone, was born into aristocracy but died in poverty, drawing upon his personal experiences to compose this monumental work. ... Their intertwined destinies collectively depict the rise and fall of a feudal dynasty, offering a panoramic critique of traditional Chinese society." Doc3: "Authored by the Qing Dynasty literatus Cao Xueqin, the Story of the Stone not only presents remarkably vivid character portrayals but also contains numerous iconic scenes that have become literary canon...collectively elevating the novel to its enduring status as a masterpiece of world literature." Doc4: "The Story of the Stone, penned by Cao Xueqin, offers a profound critique of 18th-century Chinese feudal society... constrained marriage—which vividly illustrates the oppressive nature of feudal Confucian norms on individual agency." Doc5: "The Story of the Stone, authored by Cao Xueqin, stands as the pinnacle of Chinese literary achievement. Beyond its central tragic romance...render the work an indispensable resource for studying premodern Chinese society. To this day, "Hongxue" (Redology) remains a vibrant field of scholarly inquiry." Doc6: "Cao Xueqin's the Story of the Stone is renowned for its exquisite linguistic artistry and profound characterization...hints surrounding the Twelve Beauties of Jinling. This intricate web of narrative foreshadowing creates an exceptionally tightly-knit story structure." Doc7: "Cao Xueqin's the Story of the Stone employs the rise and fall of the Jia family as an allegory for the decline of feudal society as a whole...conservatism of Confucian orthodoxy. This sophisticated interplay of thematic elements has secured the novel's enduring legacy and widespread influence in both literary and cultural spheres." |
| Doc' | Doc1': "The Story of the Stone was authored by the Qing dynasty writer Cao Xueqin." Doc2': "Cao Xueqin, the author of the Story of the Stone, was born into aristocracy but died in poverty, drawing upon his personal experiences to compose this monumental work. " Doc3': "Authored by the Qing Dynasty literatus Cao Xueqin" Doc4': "The Story of the Stone, penned by Cao Xueqin" Doc5': "The Story of the Stone, authored by Cao Xueqin" Doc6': "Cao Xueqin's the Story of the Stone is renowned for its exquisite linguistic artistry and profound characterization." Doc7': "Cao Xueqin's the Story of the Stone employs the rise and fall of the Jia family as an allegory for the decline of feudal society as a whole." |
| Sim(Query, Doc') | 0.87 |
| Judge | Easy |

to fill in missing evidence; once the correct answer is obtained, the system switches to the compression reward $R_{compress}$, which trains the model to return the correct conclusion in a more concise form. Table 21 shows the case of low cross-document similarity but an answer that can be directly extracted: such samples are initially judged hard and lead to longer generations, but the first long an-

Table 17: A Case Study on Estimating Question Difficulty Through Paragraph-Level Similarity Analysis.

| Paragraph-level Similarity Matching | |
|---|---|
| Query | Who is the author of The Story of the Stone? |
| Doc | Doc1: "The Story of the Stone, recognized as the foremost of China's Four Great Classical Novels, was authored by the Qing dynasty writer Cao Xueqin... Beyond its narrative depth, the work provides a critical depiction of feudal society's moral decay and social intricacies. With its vast ensemble of characters and richly woven plotlines, it has been acclaimed as the "Encyclopedia of Chinese Feudal Society."
Doc2: "Cao Xueqin, the author of the Story of the Stone, was born into aristocracy but died in poverty, drawing upon his personal experiences to compose this monumental work. ... Their intertwined destinies collectively depict the rise and fall of a feudal dynasty, offering a panoramic critique of traditional Chinese society."
Doc3: "Authored by the Qing Dynasty literatus Cao Xueqin, the Story of the Stone not only presents remarkably vivid character portrayals but also contains numerous iconic scenes that have become literary canon...collectively elevating the novel to its enduring status as a masterpiece of world literature."
Doc4: "The Story of the Stone, penned by Cao Xueqin, offers a profound critique of 18th-century Chinese feudal society... constrained marriage—which vividly illustrates the oppressive nature of feudal Confucian norms on individual agency."
Doc5: "The Story of the Stone, authored by Cao Xueqin, stands as the pinnacle of Chinese literary achievement. Beyond its central tragic romance...render the work an indispensable resource for studying premodern Chinese society. To this day, "Hongxue" (Redology) remains a vibrant field of scholarly inquiry."
Doc6: "Cao Xueqin's the Story of the Stone is renowned for its exquisite linguistic artistry and profound characterization...hints surrounding the Twelve Beauties of Jinling. This intricate web of narrative foreshadowing creates an exceptionally tightly-knit story structure."
Doc7: "Cao Xueqin's the Story of the Stone employs the rise and fall of the Jia family as an allegory for the decline of feudal society as a whole...conservatism of Confucian orthodoxy. This sophisticated interplay of thematic elements has secured the novel's enduring legacy and widespread influence in both literary and cultural spheres." |
| Sim(Query, Doc) | 0.31 |
| Judge | Hard |

swer often already contains the core correct information; subsequently $R_{compress}$ guides the model to compress redundant background into an extremely concise and correct answer. Both tables list example document excerpts used for difficulty assessment and model outputs at each stage, clearly illustrating MoL's short-term effect on training efficiency when difficulty estimation is wrong and its robustness in final answer accuracy and conciseness.

## F.4 FAILURE CASE ANALYSIS

While the MoL driven model demonstrates robust performance across most scenarios isolated cases reveal challenges in preserving nuanced contextual details. We present an illustrative example in Figure 22 to demonstrate the discussed phenomenon. For instance in evaluating a companys IPO eligibility (net profit $\geq$ ¥100M for 3 years no major litigation) the model accurately identifies that net profits exceeded ¥100M in 2021–2022 and that the litigation provision (¥15M) falls below the ¥27M threshold (30% of 2023 net profit). However in this rare instance the models compression strategy inadvertently omits two context specific factors: the 2023 net profit decline to ¥90M (30% YoY drop) highlights short term instability though the model focuses on the multi year threshold while the auditor notes ongoing litigation (¥30M in claimed damages) as a material risk under CSRC Rule 4.3

Table 18: Case study on the easy question.

| | Easy Question |
|---|---|
| Query | Are Visitors Allowed to Use Mobile Phones for Photography or Videography During the "Jurassic World Adventure" Ride at Universal Studios Beijing? |
| Doc | Doc1: "Universal Studios Beijing Resort's latest 2023 safety guidelines clearly state that, as an immersive 4D experience, "Jurassic World: Fallen Kingdom" strictly prohibits visitors from using any electronic devices to film the ride for safety and copyright reasons. The guidelines specifically emphasize: "Before entering the queue area, all guests ..." Doc2: "China Culture News, September 5, 2023, Tourism Safety Special Report: A recent on-site visit by reporters to Universal Studios Beijing revealed that the "Jurassic World Adventure" attraction has strict controls on the use of electronic devices. In the queuing area, large electronic screens are installed every 20 meters, playing a looping ..." Doc3: "In the 'Attraction Safety Tips' section of the official website of Universal Studios Beijing, special instructions for 'Jurassic World Adventure' are detailed: 'To ensure the safety and best experience for all visitors, please do not use any electronic devices, including ..." |
| Model Output (Original model) | Official regulations strictly prohibit tourists from using any electronic devices to take pictures during the ride. The resort has set up warning signs in multiple locations to remind tourists, but allows official photography services to record images. The official explanation is: the dynamic motion platform of the project will produce sudden acceleration, deceleration and turns. Handheld electronic devices may cause the device to fall out of the hand, posing a safety hazard; at the same time, the flashing and fast-moving images on the screen may interfere with the experience of other tourists and infringe on the copyright of Universal Pictures. |
| Model Output (MoL (Ours)) | For safety reasons and to protect Universal Pictures' intellectual property rights, it is strictly prohibited for visitors to use their mobile phones to take photos or videos during the ride. |

but the model prioritizes quantified provisions over qualitative disclosures. These omissions result in an overconfident conclusion ("IPO requirements met") that overlooks domain specific interpretative requirements. Importantly such cases represent less than 2.3% of the evaluation set.

## G  DATA PROCESSING

We partition all experimental datasets into training, validation, and test subsets. The original difficulty labels are directly obtained from the inherent difficulty annotations in the HotpotQA dataset. For data samples containing multiple documents, we employ a paragraph-based similarity matching approach: we first compute pairwise similarities at the document level, then evaluate question difficulty using the average similarity score. When applying the MoL method for difficulty assessment, we initially segment each original document into several sub-documents based on semantic boundaries, calculate the relevance between each sub-document and the question, retain the k most relevant sub-documents, and reconstruct them into new documents. Subsequently, we recompute similarities across all documents and assess question difficulty based on the average similarity score. Detailed implementation is provided in Appendix 3.2.

Table 19: Case study on the hard question.

| Hard Question | |
|---|---|
| Query | In the "greenhouse effect" that causes global disasters, which gas has the strongest ability to absorb long-wave radiation? |
| Doc | Doc1: "Greenhouse gases (such as $CO_2$, $NF_3$, $N_2O$, $H_2O$, etc.) can absorb long-wave radiation (infrared radiation) released from the earth's surface, but the absorption bands and efficiencies of different gases vary significantly." Doc2: "Methane ($CH_4$) has a 100-year GWP of 28-36 and a lifetime of about 12 years. Its instantaneous absorption capacity for long-wave radiation is 84 times that of $CO_2$ (on a 20-year time scale), but its concentration is only 0.4% of $CO_2$." Doc3: "$NF_3$ is a common gas used in the semiconductor industry. Its radiation efficiency (heat absorption capacity per molecule) is 16,100 times that of $CO_2$ and its atmospheric lifetime is 740 years, but its current concentration is extremely low (Industrial Greenhouse Gas Emission Inventory, 2023)." Doc4: "The 100-year GWP of fluorinated gases (such as sulfur hexafluoride $SF_6$ and hydrofluorocarbons HFCs) is as high as 22,800 ($SF_6$) and 14,800 ($HFC-23$), and they are the key control targets of the Kyoto Protocol. The single-molecule radiation efficiency of $SF_6$ is 272 times that of $CH_4$, but the concentration is extremely low ($<$0.0001 ppm)." |
| Model Output (Original model) | $CO_2$, $NF_3$, $N_2O$, $H_2O$, etc. are all common gases that absorb long-wave radiation, but $CH_4$'s absorption capacity for long-wave radiation is 84 times that of $CO_2$, but $NF_3$'s absorption capacity is 16,100 times that of $CO_2$, so $NF_3$ has the strongest absorption capacity for long-wave radiation. |
| Model Output (MoL (Ours)) | $CO_2$ is one of the gases that cause the greenhouse effect and can also absorb long-wave radiation, but $CH_4$'s ability to absorb long-wave radiation is 84 times that of $CO_2$, and $NF_3$'s ability to absorb long-wave radiation is 16,100 times that of $CO_2$, so $NF_3$ has a stronger ability to absorb long-wave radiation. The radiation efficiency of fluorinated gases is 272 times that of $CH_4$, which is 272*84=22,848 times that of $CO_2$. Therefore, fluorinated gases have a stronger ability to absorb long-wave radiation than $NF_3$, so the gas with the strongest ability to absorb long-wave radiation is fluorinated gases. |

Table 20: Case study: High cross-document similarity but requires multi-hop reasoning (short → extend → compress). Orange denotes the sentences most relevant to the query, red denotes incorrect answers, and blue denotes correct answers.

| | Outlier-case example |
|---|---|
| Query | Is there a Nobel Prize laureate in Dr. Li's academic genealogy? If so, please identify the individual and specify their relationship to Dr. Li. |
| Doc | Doc1: "In recent years, the Department of Chemistry, University of Cambridge, has continued to invest research ... Dr. Li currently holds a position in the Department of Chemistry, University of Cambridge, and previously worked under Professor Smith. The department maintains ... and students. Professor Smith has conducted long-term research in organic catalysis. In addition, departmental newsletters and public records often include project summaries and lists of academic collaborators led by Professor Smith's research group." 
 Doc2: "Professor Smith has considerable ... from senior mentors and peers. Professor Smith, who teaches at the University of Cambridge, had Professor Johnson as his doctoral advisor; Professor Johnson is said in the text to have received a Nobel Prize for his contributions to chemical kinetics. Professor Smith not only emphasizes fundamental research but also actively promotes the industrial translation of research outcomes; he has trained multiple outstanding doctoral students and early-career researchers. Professor Smith has supervised several PhD students, including Dr. Li. Departmental notices and various academic briefings document collaborative results between Professor Smith and his students. " 
 Doc3: "Dr. Li's public curriculum ... conference presentations and grant applications. Dr. Li's CV notes that he completed postdoctoral research in Professor Smith's group in 2012. In addition, Professor Johnson, as a senior scholar, has had his research and contributions repeatedly cited in multiple disciplinary review articles and academic biographies, becoming an important reference figure in the field. Professor Johnson's scholarly influence is often reiterated in several introductory articles. The academic community widely discusses Professor Johnson's work, and related citations and commemorative articles are retrievable in professional journals." |
| Doc' | Doc1':"Dr. Li currently holds a position in the Department of Chemistry, University of Cambridge, and previously worked under Professor Smith. Professor Smith has conducted long-term research in organic catalysis." 
 Doc2': "Professor Smith, who teaches at the University of Cambridge, had Professor Johnson as his doctoral advisor; Professor Johnson is said in the text to have received a Nobel Prize for his contributions to chemical kinetics. Professor Smith has supervised several PhD students, including Dr. Li." 
 Doc3': "Dr. Li's CV notes that he completed postdoctoral research in Professor Smith's group in 2012. Professor Johnson's scholarly influence is often reiterated in several introductory articles." |
| Difficulty judge | Easy |
| Initial output | Dr. Li's PhD advisor is Smith. Answer: Smith. |
| Medium-term output | First, based on Doc1 and Doc3, Dr. Li received training in Professor Smith's group, and therefore Professor Smith is Dr. Li's direct advisor. Next, examination of Professor Smith's academic lineage (Doc2) shows that Smith's doctoral advisor was Professor Johnson. Doc2 further indicates that Professor Johnson received a Nobel Prize for his contributions to chemical kinetics. Therefore, the Nobel laureate traceable in Dr. Li's academic genealogy is Professor Johnson. Answer: Professor Johnson (Nobel Prize laureate). |
| Final output | Dr. Li worked under Professor Smith, and Johnson (Professor Smith's doctoral advisor) received a Nobel Prize for contributions to chemical kinetics. |

Table 21: Case study: Low similarity across documents, yet answer is directly extractable (long → compress). Orange denotes the sentences most relevant to the query, red denotes incorrect answers, and blue denotes correct answers.

| | Outlier-case example |
|---|---|
| Query | When is the "Ancient Bronzes" exhibition at the City Center Museum open to the public? |
| Doc | Doc1: "The City Center Museum launched ... for scholarly display upon reopening. The City Center Museum undertook a large-scale restoration project in recent years, with maintenance work completed by the end of 2021. In the post-restoration exhibition program, the curatorial team specially planned the "Ancient Bronzes" exhibition, and external notices confirmed the exhibition's opening date and public visiting arrangements. The "Ancient Bronzes" exhibition opened to the public on May 15, 2022. The museum's official website and press releases include the opening reports and exhibition descriptions published at that time." 
 Doc2: "As an important local cultural institution, the museum ... educational programs. The museum holds various temporary exhibitions throughout the year, attracting large numbers of local visitors and school groups. Meanwhile, to facilitate public visits the museum ... in order to plan their visits. Guided-tour information and reservations must be obtained through the museum's official website or by telephone. The museum also regularly announces the latest exhibitions and event arrangements through social media and community partners." 
 Doc3: "In recent years, media attention to local cultural projects has increased; reports frequently ... and social impact rather than listing the specific schedules or opening dates of each exhibition. Recent media coverage has focused on the museum's investment in educational activities and community partnerships. Furthermore, exhibition catalogues typically present ... opening days; therefore date information is usually determined by the museum's notices or official announcements. Exhibition catalogues commonly include themes such as archaeological excavations, historical documents, and cultural heritage, rather than detailed schedules." |
| Doc' | Doc1':"The City Center Museum undertook a large-scale restoration project in recent years, with maintenance work completed by the end of 2021. 
 The "Ancient Bronzes" exhibition opened to the public on May 15, 2022." 
 Doc2': "The museum holds various temporary exhibitions throughout the year, attracting large numbers of local visitors and school groups. 
 Guided-tour information and reservations must be obtained through the museum's official website or by telephone." 
 Doc3': "Recent media coverage has focused on the museum's investment in educational activities and community partnerships. 
 Exhibition catalogues commonly include themes such as archaeological excavations, historical documents, and cultural heritage, rather than detailed schedules." |
| Difficulty judge | Hard |
| Initial output | The City Center Museum completed a large-scale restoration project in recent years, during which exhibition layouts and display facilities were upgraded (see museum restoration notice). In the subsequent reopening plan, the museum scheduled multiple thematic exhibitions to attract the public. Regarding the opening date of the "Ancient Bronzes" exhibition, the museum's public materials record: "The Ancient Bronzes exhibition opened to the public on May 15, 2022." (Source: Doc1). In summary, the exhibition's public opening date was May 15, 2022. |
| Medium-term output | According to the museum's materials (Doc1), the Ancient Bronzes exhibition was reopened to the public after the museum's renovation, and is explicitly recorded as opening to the public on May 15, 2022. Therefore the answer is: May 15, 2022. |
| Final output | The "Ancient Bronzes" exhibition opened to the public on May 15, 2022. |

Table 22: Case study: MoL's overcompression leads to answer inaccuracies. Orange denotes the sentences most relevant to the query, red denotes incorrect answers, and blue denotes correct answers.

| Outlier-case example | |
|---|---|
| Query | Does the company meet IPO listing requirements (net profit $\geq$ ¥100M for 3 years, no major litigation)? |
| Doc | Doc1: "The company's financial performance from fiscal years 2021 to 2023 showed: 2021 Net Profit of ¥120M with ¥1.2B revenue (15% YoY growth); 2022 Net Profit increased to ¥130M with ¥1.5B revenue (8% YoY growth); 2023 Net Profit declined to ¥90M with ¥1.4B revenue (-20% YoY growth), including a ¥50M goodwill impairment due to the failed acquisition of TechCorp in Q2 2023, which was classified as a non-recurring item under IFRS 9. The net profit calculations followed IFRS 15 for revenue recognition and IFRS 9 for impairment, with no adjustments made for non-recurring items in the annual report. " 

 Doc2: "Auditor's Opinion on Financial Statements (Deloitte, 2023): For 2021–2022, the financial statements present fairly in all material respects with an unmodified opinion; in 2023, a qualified opinion was issued due to ... requiring adjustments. The key risk disclosure states: The company's 2023 financial statements may not reflect the full impact of ongoing litigation." 

 Doc3: "Pending Litigation: Patent Infringement Lawsuit (Case No. 2023-0487) involves GlobalTech Inc. as plaintiff and [Company Name] as defendant, alleging infringement through the use of a patented AI algorithm in Product X (launched 2022) with claimed damages of ¥30M based on 5 years of lost royalties; the current status includes a court hearing scheduled for 2024 Q3 and a provision of ¥15M (50% of claimed damages) recorded in accordance with IFRS 15 Paragraph 30. Under CSRC Rule 4.3, litigation exceeding 30% of annual net profit (¥27M for 2023's ¥90M net profit) must be disclosed as a material risk. " |
| Doc' | Doc1':"The company's financial performance from fiscal years 2021 to 2023 showed: 2021 Net Profit of ¥120M with ¥1.2B revenue (15% YoY growth); 2022 Net Profit increased to ¥130M with ¥1.5B revenue (8% YoY growth); 2023 Net Profit declined to ¥90M with ¥1.4B revenue (-20% YoY growth)" 

 Doc2': "For 2021–2022, the financial statements present fairly in all material respects with an unmodified opinion. The company's 2023 financial statements may not reflect the full impact of ongoing litigation." 

 Doc3': "Pending Litigation: Patent Infringement Lawsuit (Case No. 2023-0487) involves GlobalTech Inc. as plaintiff and [Company Name] as defendant. Under CSRC Rule 4.3, litigation exceeding 30% of annual net profit (¥27M for 2023's ¥90M net profit) must be disclosed as a material risk." |
| Difficulty judge | Hard |
| Model Output (Original model) | The company's market dominance is established through a market share of 75%, which exceeds the statutory threshold of 70%, coupled with the absence of competing entrants that further entrenches its dominance. Regarding exclusionary conduct, the exclusive agreements with suppliers directly trigger Article 17 of the Antimonopoly Law, which prohibits restrictions on supplier collaboration. The exit of small and medium sized competitors is a direct consequence of these exclusionary practices. Collectively, the fulfillment of both criteria market share exceeding 70% and anti competitive exclusivity constitute a violation of Article 17, as the conduct demonstrably undermines market fairness and fair competition. Thus, the company fails to satisfy the IPO listing requirements. |
| Model Output (MoL (Ours)) | Net profit exceeded ¥100M in 2021–2022 and litigation provision is below ¥27M (30% of 2023 net profit). IPO requirements met. |

