# OpenReview forum: "MoL: Adaptive Mixture-of-Length Reasoning for Efficient Question Answering with Context"
_ICLR.cc/2026/Conference — ICLR 2026 Poster_

### Official Review · Reviewer_UwkB · 2025-10-30

**Soundness:** 2
**Presentation:** 3
**Contribution:** 3
**Rating:** 6
**Confidence:** 3

**Summary:**

The paper proposes a training framework called Mixture-of-Length (MoL). It produces short answers on easy questions and expand reasoning when the question is hard. The approach estimates task difficulty using average cross-document similarity and utilizing set-cover view of information redundancy across documents. They also train an LLM in GRPO-style with a dual reward that switches between a compression reward for correct answers and an extension reward for incorrect ones, with curriculum-style scheduling of the length coefficient. Experiments on HotpotQA, StrategyQA, and Loong datasets across several base LLMs show higher accuracy with fewer tokens.

**Strengths:**

- The problem tackled in the paper is important and well-formulated.
- The set-cover motivation and its practical proxy via average cross-document cosine similarity are clear and intuitive.
- Table 1 shows that accuracy increases with meaningful token reductions for several base LLMs, trained by the author's method.
- The paper is clear about limitations, ethics, and contains all needed information for reproducibility.

**Weaknesses:**

- Equating cross-document similarity with "easy" questions risks misclassifying questions that demand nontrivial reasoning over redundant sources. Note that authors treat a question as easy if its source documents look "similar", which they estimate by taking a few relevant sentences from each document and averaging how alike their embeddings are. This approach may work well on the evaluated datasets, but its generalizability to real-world scenarios remains uncertain. For example, imagine 10 near-duplicate news articles about a merger. A question like "By how many quarters did revenue grow between the two pre-merger reports?" requires number extraction, comparison, and arithmetic, despite the fact that the documents are very similar. In this case, high similarity means that their proxy call the question "easy", but the reasoning to answer the question correctly is non-trivial.
- Claims of "statistically significant" gains appear, but I did not find details on tests, confidence intervals, or seeds across runs.
- The paper compares "Tokens" across models and methods, but the tokenization standard (e.g., tokenizer family) and measurement point (prompt+answer? answer only?) are not specified; comparability is unclear when base models differ.

Minor issue:
- The text on some Figures (especially Fig. 5) is very small and hard to read.

**Questions:**

1. I would suggest a stress-test for your method to decouple document redundancy from reasoning depth, required to answer the question, using two new datasets:

- High-similarity + high-reasoning: near-duplicate docs but questions require multi-step computation/logic across them.
- Low-similarity + low-reasoning: stylistically diverse docs where each answer is a direct lookup.

If I understood the paper correctly, your proxy should call the first set "easy", and the second "hard", despite that the reality is the opposite. How will your method work in this situation?

2. At lines 209-212 you write: "The specific length thresholds are empirically set based on the response length distribution observed in the HotpotQA dataset". Can you please elaborate?

3. Why there is a "2" multiplier in the Formula 4?

4. The Figure 2 is hard to understand, since it doesn't really match with the text. What are these "compress" and "extend" stages, depicted on the Figure 2? In the text itself, I only see the description of the compression reward, not the distinct stage called like this. Can you please explain?

---

> ### Author Response · Authors · 2025-11-21
> **Response to Reviewer UwkB (1/2)**
>
> We thank the reviewer for the detailed review of the paper and the valuable feedback. Below, we address the reviewer's questions in a point-by-point manner.
> ***
> **W1: Flawed Difficulty Metric & Q1: Stress-Test Suggestion**
>
> We thank the reviewer for the insightful suggestion. The reviewer is correct that our similarity based heuristic can misclassify difficulty in the two proposed scenarios. We empirically validate this observation through controlled experiments and present illustrative examples in Appendix F.3.
>
> The test results indicate that the framework is robust to such misclassifications thanks to a self-correcting dynamic driven by answer correctness rather than the initial difficulty label. In cases of high similarity and high reasoning, the heuristic tends to label the example as easy and set $L_{target}$ to a short value. The model may initially produce a concise but incorrect answer, which triggers the $R_{extend}$ reward and penalizes overly short responses. Over training the model learns that a longer step by step reasoning chain is required for correctness and higher return, effectively compensating for the initial easy label. In cases of low similarity and low reasoning, the heuristic tends to label the example as hard and set $L_{target}$ to a long value. The model may produce a lengthy yet correct answer via simple lookup, thereby triggering the $R_{compress}$ reward and encouraging reduced verbosity. As training progresses, the model moves beyond the constraints of the initial target lengths and learns to identify optimal response patterns.
>
> In summary, incorrect difficulty estimates mainly affect training efficiency rather than the correctness of the final learned policy. We have added this clarification to the Section B.5 to better explain the resilience of our framework.
> ***
> **W2: "Statistically significant" claims are unsubstantiated.**
>
> We apologize for the confusion. The statement was meant to emphasize that the trained model substantially outperforms the base model. We have revised the wording accordingly in the manuscript.
> ***
> **W3: "Tokens" are not well-defined.**
>
> We thank the reviewer for pointing out this ambiguity. Action taken: we updated Section 4, Experimental Settings, to state explicitly that the term "tokens" refers to the length of the generated response and counts only answer tokens.
> ***
> **Minor issue: Figure 5 text is very small.**
>
> Thank you for pointing this out. In the final version we will enlarge the font size of all numeric labels, with particular attention to Figure 5, to improve readability.
> ***
> **Q2: Elaboration on empirical length thresholds.**
>
> Thank you for the suggestion. We conducted a systematic analysis on the first 1,000 samples from the HotpotQA dataset. First, We use the Qwen3-8B model to directly classify the difficulty of questions. Subsequently, we analyzed the distribution of output lengths generated by the model for each difficulty tier (Figure 6). The average output lengths of the model for each difficulty tier are summarized in the following result:
>
> |Difficulty|Simple|Medium|Hard|
> |:--------|:--------|:--------|:--------|
> |Tokens length (Average)|441|897|2003|
>
> Through statistical analysis of these distributions, we determined the target lengths as $L_{target}$={512,1024,2048}, corresponding to simple, medium, and complex questions respectively. This data-driven approach ensures that the thresholds align with the inherent reasoning complexity while maintaining computational efficiency.
>
> To validate the impact of $L_{target}$ on the experimental results, we also conducted additional experiments (Figure 7 & Figure 8). We found that $L_{target}$ primarily affects training efficiency, while having minimal influence on the final performance. Even when the target length is imperfectly specified, the model can still gradually discover an appropriate response pattern during training in an adaptive manner through MoL.
> ***
> **Q3: Why there is a "2" multiplier in the Formula 4?**
>
> We apologize for the confusion. The factor of two in the numerator arises from averaging pairwise similarities between documents. With $n$ documents there are $\frac{n \cdot (n - 1)}{2}$ unordered pairs. If $S$ denotes the sum of all pairwise similarities over i < j, the average pairwise similarity is $S$ divided by $\frac{n \cdot (n - 1)}{2}$, which is algebraically equal to $\frac{2 \cdot S}{n \cdot (n - 1)}$. This algebraic rearrangement explains the factor of two in Equation 4.

---

> ### Author Response · Authors · 2025-11-21
> **Response to Reviewer UwkB (2/2)**
>
> **Q4: Figure 2 is hard to understand.**
>
> Thank you for the feedback. The "compress" and "extend" stages in Figure 2 represent dynamic reward modes (not sequential phases) governed by the dual-objective reward (Section 3.3):
> - Compress: Activated for correct answers to enforce brevity (via $R_{compress}$​).
> - Extend: Triggered for errors to enable error recovery through longer reasoning (via $R_{extend}$​).
>
> When a question is answered incorrectly, the model is encouraged by $R_{extend}$ to lengthen its response in order to search for missing evidence chains, whereas once the question is answered correctly, $R_{compress}$ rewards more concise expressions. These two behaviors are adaptively interleaved during training based on the correctness of the current response, rather than forming a fixed multi-stage pipeline. We apologize for the confusion and we will update Figure 2’s caption to clarify it depicts training-time adaptive reward modulation.
> ***
> Thank you again for your insightful feedback, that helped us improve our work.

---

### Official Review · Reviewer_bLCr · 2025-10-31

**Soundness:** 3
**Presentation:** 3
**Contribution:** 3
**Rating:** 6
**Confidence:** 3

**Summary:**

This paper introduces Mixture-of-Length (MoL), a reinforcement learning framework for question answering that adaptively controls response length based on question difficulty. Difficulty is estimated via an information-theoretic proxy: inter-document redundancy computed as the average cosine similarity of top-k question-relevant sentences (low similarity ⇒ harder).
MoL couples this with a dual reward scheme: correct answers receive a compression reward encouraging brevity, while incorrect ones get an extension reward promoting longer reasoning; a progressive schedule anneals the length coefficient over training.
Across three datasets, MoL improves accuracy while reducing output tokens. The approach generalizes to unseen QA datasets, and a post-hoc activation analysis suggests adaptive computation (fewer active layers on easy questions).

**Strengths:**

- The use of an information-theoretic proxy rooted in the set cover problem for assessing question difficulty is a solid theoretical motivation.
- The dual-objective reward mechanism of MoL (compression and expansion) is well-formulated and empirically separated in ablations.
- The paper presents rigorous results across three QA datasets, as well as additional analyses and ablation studies that improve the reliability of central claims.
- MoL’s response-length adaptation is shown empirically with fewer tokens for easy and more for hard questions, outperforming strong baselines.

**Weaknesses:**

- Although Table 2 compares with passage-based similarity and HotpotQA labels, the difficulty prediction step relies on a Top-k sentence selection followed by embedding similarity. It is unclear how sensitive this is to k, embedding model choice, or sentence granularity. More systematically controlled ablation studies are needed.
- The paper shows strong overall gains, but lacks qualitative or quantitative analysis of scenarios where MoL fails — e.g., cases where over-compression leads to loss of crucial reasoning steps, especially in complex or long-context queries. Identifying such cases would strengthen the empirical claims and help users understand the trade-offs of the method.
- The analysis showing a correlation between shorter generated outputs and fewer activated layers is interesting, but the work does not demonstrate causal influence. It remains unclear whether the reduced computation is a deliberate model behavior or merely a post-hoc artifact of the compression objective.
- No human evaluation of response quality.

**Questions:**

1. On Difficulty Estimation: How sensitive is your difficulty metric to the choice of k (Top-k sentence filtering), the embedding model, and the document segmentation strategy? Did you experiment with alternative encoders or granularities?
2. Error/Failure Cases: Can you provide examples where MoL over-compresses and fails to capture necessary reasoning steps (e.g. in complex multi-hop or long-context queries)? How often does this occur in practice, and are there mechanisms to detect or mitigate such failures?
3. Internal Activation Adaptation: Have you tried any controlled intervention experiments (e.g., freezing certain layers) to test whether the reduced activation in “easy” cases is causally induced by MoL, rather than being an emergent side effect of training?
4. Human Evaluation: Did you consider running a human evaluation on a subset of responses to measure perceived quality and readability? This would be especially valuable given the strong emphasis on token compression — are the shorter outputs still informative to end users?
5. Generalizability Across Tasks: Do you expect the MoL reward framework to extend effectively to other domains (e.g., dialogue reasoning, code generation, or fact verification)? What adaptations (if any) would be required?

---

> ### Author Response · Authors · 2025-11-21
> **Response to Reviewer bLCr (1/1)**
>
> We thank the reviewer for the detailed review of the paper and the valuable feedback. Below, we address the reviewer's questions in a point-by-point manner.
> ***
> **W1 & Q1:Sensitivity of Difficulty Estimation**
>
> We performed ablation studies on the choice of k (Top-k), the embedding model, and the sentence granularity. All experiments used the Qwen3-8B model on the Loong dataset. Below we report the ablation on k values:
>
> |Method|Loong (Acc)|Loong (Tokens)|
> |:--------|:--------|:--------|
> |MoL (k=1)|61.7|**1199**|
> |MoL (k=2)|**62.3**|1374|
> |MoL (k=3)|62.1|1422|
>
> To evaluate the impact of the embedding model on experimental performance, we conduct comparative experiments using Sentence-T5. The results are summarized below:
>
> |Method|Loong (Acc)|Loong (Tokens)|
> |:--------|:--------|:--------|
> |MoL (Sentence-T5)|**62.6**|1462|
> |MoL (BGE-M3)|62.3|**1374**|
>
> We conducted experiments under different sentence granularities by treating either a single sentence or a pair of consecutive sentences as the basic unit. The experimental results are reported below:
>
> |Method|Loong (Acc)|Loong (Tokens)|
> |:--------|:--------|:--------|
> |MoL (one sentence)|62.3|**1374**|
> |MoL (two sentence)|**62.5**|1439|
>
> Our experiments show that the proposed method is highly robust to the choice of k, the embedding model, and the sentence granularity. This robustness arises from the stability of our difficulty estimation.
>
> We thank the reviewer for the insightful suggestion, which has significantly strengthened the rigor of our ablation study. The corresponding experiments are now incorporated in Appendix B.7.
> ***
> **W2 & Q2:Failure Case Analysis**
>
> We sincerely thank the reviewer for this insightful comment. In response, we have added a detailed analysis of failure cases in Appendix F.4, focusing on scenarios where MoL’s compression mechanism inadvertently removes critical reasoning steps. Importantly such cases represent less than 1.3% of the evaluation set. This is achieved by our bidirectional reward function which reinvokes $R_{extend}$​ when over compression leads to incorrect answers encouraging the model to expand its reasoning to recover the missing evidence chain. This mechanism significantly reduces the probability of such errors occurring in practice.
> ***
> **W3 & Q3: Correlational Nature of Activation Analysis and Causal Influence**
>
> The reviewer’s observation is correct. Our analysis is indeed post hoc and correlational, and cannot definitively establish causality. However, we believe that the change in the number of activated layers is not merely a byproduct of the compression objective. After training with MoL, the model exhibits a higher average number of activated layers when answering difficult questions compared to the pre-training baseline, while simultaneously producing more concise outputs (i.e., shorter token sequences). This observation contradicts the view that the observed changes are merely a post-hoc artifact of the compression objective. Instead, we believe that, after training, the model learns to adaptively select layers for reasoning: for simple questions, only a small number of layers are needed, whereas more complex questions require deeper layer utilization.
> ***
> **W4 & Q4: Lack of Human Evaluation**
>
> We thank the reviewer for this insightful suggestion. To address it, we conducted a human evaluation on 100 randomly sampled compressed outputs from the dataset. For each question, domain experts were shown both the compressed response produced by the MoL trained model and the corresponding response from the pretraining baseline, and were asked to assess readability and information sufficiency. They found that 94 percent of the compressed outputs were overall better than the pretraining baseline. We have added a selection of representative examples to Appendix F.
> ***
> **Q5: Generalizability Across Tasks**
>
> We thank the reviewer for raising this important point about generalizability. Our bidirectional reward framework is inherently task agnostic as it learns to balance token level and answer level objectives without relying on domain specific heuristics. This is evidenced by its effectiveness in SVAMP dataset and  CommonsenseQA dataset (Table 5; QA without reference documents), where it dynamically adjusts generation length by comparing predicted rewards against reference answers. For adaptation to other domains, the framework allows plugging in domain specific reward shaping: for dialogue reasoning, coherence metrics could prioritize response relevance; for code generation, structural constraints could guide $R_{extend}$​; and for fact verification, target lengths could scale with evidence complexity. The core assumption is access to reference answers or quality signals (e.g., test case coverage) though weak supervision (e.g., human preferences) could substitute in open ended tasks.
> ***
> Thank you again for your insightful feedback, that helped us improve our work.

---

### Official Review · Reviewer_V4Cn · 2025-11-01

**Soundness:** 2
**Presentation:** 2
**Contribution:** 2
**Rating:** 4
**Confidence:** 3

**Summary:**

This work proposes a reward function for reinforcement learning to improve the reasoning capability of large language models by adaptively controlling the length of generation based on the difficulty assessment. Basic idea comes from the prior studies in selecting the longer length of generation especially for complex reasoning tasks demanding longer inference, but shorter to avoid spurious reasoning. This work leverages the difficulty of a task measured by the similarity of retrieved sentences, i.e., the more difficult when almost no relevant sentences are extracted, and used it to adaptively choose the generation length heuristically. Experiments show that proposed approach achieves improvements when compared with related baselines.

**Strengths:**

- The proposed approach defines an interesting reward function to adaptively control the length of generation based on the difficulty of a task. The difficulty is measured by the similarity of retrieved sentences, such that the more similar implies the more easy to solve a task.
- Experiments are carried out systematically demonstrating gains on standard benchmarks when compared with other related methodologies. Further analyses show the generalization capacity over different domains and its relation of the difficulty assessment and accuracy.

**Weaknesses:**

- It is not clear whether the difficulty level could be attributed solely by the similarity of retrieved answer or not. There might be a case of a query which has a direct answer in a single sentence, and no similar sentences exist in the document pool. In this case, the task is treated as an extremely difficult instance leading to over estimates. I think it is better to show whether the difficulty assessment is correlated with the actual difficulty or not. It is possible to measure the accuracies in general by existing models or some evidence, e.g., the distribution of relevant passages in the document pool.
- The motivation of the heuristics to map the difficulty level to its target length is not clear. Probably it was judged empirically, but the more clear discussion will be necessary, e.g., the plot of the actual difficulty level of a task and its length for generation.
- Math notations are not consistent and several symbols are not defined. For example, $S_i$ is used as a subset of a document but $S_{ij}$ is defined as a scalar for similarity. $s$, Top-k and $Sim$ are not defined in Equation 2.

**Questions:**

See weaknesses.

---

> ### Author Response · Authors · 2025-11-21
> **Response to Reviewer V4Cn (1/2)**
>
> We thank the reviewer for the detailed review of the paper and the valuable feedback. Below, we address the reviewer's questions in a point-by-point manner.
> ***
> >It is not clear whether the difficulty level could be attributed solely by the similarity of retrieved answer or not. There might be a case of a query which has a direct answer in a single sentence, and no similar sentences exist in the document pool. In this case, the task is treated as an extremely difficult instance leading to over estimates. I think it is better to show whether the difficulty assessment is correlated with the actual difficulty or not. It is possible to measure the accuracies in general by existing models or some evidence, e.g., the distribution of relevant passages in the document pool.
>
> Firstly, how to define the actual difficulity? We posit that models are more likely to answer simpler questions correctly, whereas their accuracy decreases as question difficulty increases. Consequently, we evaluate the quality of different difficulty partitioning methods by examining the accuracy difference across difficulty levels. A partitioning scheme that yields a larger accuracy difference between levels is regarded as a more reliable and reasonable difficulty categorization.
>
> So we first employ the DeepSeek-V3 model to perform difficulty categorization on the HotpotQA dataset, partitioning it into Simple and Hard subsets. Subsequently, we apply the MoL framework to re-categorize the dataset, revealing a 63.2% agreement rate between the two methods. Finally, we evaluate the Qwen3-8B model’s accuracy on both partitioned subsets. The results are summarized below:
>
> |Method|Simple (Acc)|Hard (Acc)|Difference|
> |:--------|:--------|:--------|:--------|
> |Prompt(DS-V3)|64.1|55.7|8.4|
> |MoL(Ours)|**79.2(higher)**|**50.1 (lower)**|**29.1**|
>
> The experimental results demonstrate that accuracy differences between subsets derived directly from model based categorization remain minimal (8.4%). However the MoL partitioned subsets exhibit a significant 29.1% accuracy gap confirming the unreliability of direct model driven difficulty partitioning. In contrast the MoL based difficulty assessment method proves more effective.
>
> Furthermore, We provide examples of misclassified question difficulty in Appendix F.3 (including the case you mentioned). Under our design, the length rewards ($R_{compress}$ $/$ $R_{extend}$) are jointly determined by the difficulty assessment and the correctness of the model’s response. Even when the difficulty of a question is overestimated, we still condition the decision to compress or extend the response on whether the answer is correct. This allows the model to adaptively receive length rewards and thereby converge toward an optimal response pattern.
> ***
> >The motivation of the heuristics to map the difficulty level to its target length is not clear. Probably it was judged empirically, but the more clear discussion will be necessary, e.g., the plot of the actual difficulty level of a task and its length for generation.
>
> Thank you for the suggestion. We conducted a systematic analysis on the first 1,000 samples from the HotpotQA dataset. First, We use the Qwen3-8B model to directly classify the difficulty of questions. Subsequently, we analyzed the distribution of output lengths generated by the model for each difficulty tier(Figure 6). The average output lengths of the model for each difficulty tier are summarized in the following result:
>
> |Difficulty|Simple|Medium|Hard|
> |:--------|:--------|:--------|:--------|
> |Tokens length (Average)|441|897|2003|
>
> Through statistical analysis of these distributions, we empirically determined the target lengths as $L_{target}$={512,1024,2048}, corresponding to simple, medium, and complex questions respectively. This data-driven approach ensures that the thresholds align with the inherent reasoning complexity while maintaining computational efficiency.
>
> To validate the impact of $L_{target}$ on the experimental results, we also conducted additional experiments (Figure 7 & Figure 8). We found that $L_{target}$ primarily affects training efficiency, while having minimal influence on the final performance. Even when the target length is imperfectly specified, the model can still gradually discover an appropriate response pattern during training in an adaptive manner through MoL.

---

> > ### Comment · Reviewer_V4Cn · 2025-11-27
> >
> > Thank you for the additional experiments.
> >
> > It seems to have changes in the final score, but it is not clear whether the difficulty level, length and similarities are correlated so that the similarity could be a proxy for the judgement.

---

> > > ### Author Response · Authors · 2025-11-27
> > >
> > > We thank the reviewer for their valuable feedback. To ensure precise analysis, we first partitioned the HotpotQA dataset into 100 intervals stratified by similarity and evaluated the Qwen3-8B model's accuracy on each interval. For presentation clarity, we aggregated these 100 intervals into ten broader groups (each containing ten consecutive intervals) to summarize trends in the table below:
> > >
> > > |Similarity|Acc|Tokens|
> > > |:--------|:--------|:--------|
> > > |[0, 0.1]|47.2|2501|
> > > |(0.1, 0.2]|50.1|2216|
> > > |(0.2, 0.3]|52.7|1784|
> > > |(0.3, 0.4]|57.4|1435|
> > > |(0.4, 0.5]|64.2|1011|
> > > |(0.5, 0.6]|63.7|746|
> > > |(0.6, 0.7]|64.6|613|
> > > |(0.7, 0.8]|69.4|548|
> > > |(0.8, 0.9]|78.9|407|
> > > |(0.9, 1.0]|82.8|376|
> > >
> > > Separately, we conducted a Spearman’s rank correlation analysis directly on the 100 intervals stratified by similarity to quantitatively validate the relationship between similarity, task difficulty (reflected by model accuracy), and response length. The results demonstrate that the Spearman’s ρ between similarity and accuracy is 0.91, while the correlation between similarity and token length is −0.93. These findings statistically confirm that similarity serves as a valid basis for evaluation.
> > >
> > > We would like to express our sincere gratitude for your insightful feedback. We hope this response adequately addresses your concerns.

---

> ### Author Response · Authors · 2025-11-21
> **Response to Reviewer V4Cn (2/2)**
>
> >Math notations are not consistent and several symbols are not defined. For example,  is used as a subset of a document but  is defined as a scalar for similarity. , Top-k and  are not defined in Equation 2.
>
> We thank the reviewer for the careful reading and for pointing out these regrettable errors. We have thoroughly revised Section 3.2 (lines 170-172) of the updated manuscript to correct them.
>
> $Sim(s, q)$ denotes the cosine similarity between the embeddings of sentence $s$ and question $q$.
>
> $Top$-$k(·)$ denotes the operator that returns the set of k sentences with the highest similarity scores.
> ***
> Thank you again for your insightful feedback, that helped us improve our work.

---

### Official Review · Reviewer_Wcic · 2025-11-03

**Soundness:** 3
**Presentation:** 3
**Contribution:** 3
**Rating:** 6
**Confidence:** 3

**Summary:**

The paper addresses the trade-off between reasoning quality and response efficiency in open-domain question answering with context. Traditional approaches often just compress all answers uniformly (risking under-explaining complex queries). This work proposes Mixture-of-Length (MoL), a framework to achieve ``intelligent brevity'': short answers for easy questions and longer reasoning for hard ones.

To achieve this property the authors proposed the following approaches. First, a question complexity via cross-document redundancy is estimated. Intuitively, if multiple context documents share overlapping information, the question is likely simple (high redundancy means low complexity), whereas if each document contributes unique facts (low redundancy), the question requires complex multi-hop reasoning.

Second, a dual-objective reward mechanism (trained via reinforcement learning) adaptively modulates answer length. The model is rewarded for conciseness when correct and for elaborating more when the initial answer is incorrect. Concretely, MoL defines a compress reward for correct answers that encourages minimal sufficient explanation, and an extend reward for incorrect answers that encourages providing a longer, more detailed reasoning chain. The target answer length is dynamically set based on the estimated difficulty.

Empirical validation on three benchmarks - HotpotQA (multi-hop QA), StrategyQA (implicit reasoning QA), and Loong (long-document QA) – showing MoL achieves competitive or better accuracy with dramatically fewer tokens than baselines.

**Strengths:**

1) The proposed MoL framework intelligently balances brevity and thoroughness. Unlike prior methods, MoL's dual-objective approach allows it to produce concise answers for easy questions and expand reasoning for harder ones as needed.

2) By measuring cross-document similarity after extracting key sentences, the proposed difficulty metric can reliably distinguish simple retrieval questions (high redundancy across documents) from complex multi-hop ones (low redundancy).

3) MoL achieves significant token reduction without sacrificing accuracy, often even improving it. For example, with a Qwen-8B model on HotpotQA, MoL used ~50% fewer tokens than the original model yet boosted accuracy by +6.2 points.

4) Extensive experimental evaluation. The authors considered multiple base LLMs across three diverse QA benchmarks. They compare MoL with strong baselines and MoL outperforms them. The paper also includes an ablation study isolating the impact of each component. This comprehensive evaluation strengthens confidence in the results.

5) The paper is well-written and clearly explains the intuition behind MoL.

**Weaknesses:**

1) MoL approach may still struggle on cases where complexity isn't captured by document redundancy (for instance, a single-document question that requires complex logical reasoning or implicit knowledge). The method is currently tailored to multi-document scenarios, which might limit its direct applicability to single-document QA or other formats without further adaptation.

2) The framework relies on several hand-tuned thresholds and parameters that may need adjustment in new settings. It looks like prior choices of these hyper-parameters are somewhat arbitrary and task-specific. It’s not fully tested how sensitive the approach is to these values – a different domain might require re-calibration. The authors do include an ablation on some of these hyper parameters, but the complexity of tuning so many parameters could be a drawback.

3) The solution uses reinforcement learning approach GRPO which is known to be difficult to train and resource-intensive. The experiments used a high-end setup (64× A100 GPUs), indicating significant compute requirements to reproduce or fine-tune MoL. This raises concerns about accessibility and stability of the method – researchers or practitioners with less resource might find it hard to apply.

4) The evaluation focuses on F1 accuracy and token counts, which is appropriate, but it would be interesting to know if the conciseness achieved by MoL is indeed judged favorably by humans.

5) The paper reports a fascinating correlation between question difficulty, answer length, and the number of transformer layers effectively used. However, the authors acknowledge this analysis is post-hoc and correlational, not proving a causal link. This is not exactly a flaw in the method, but a limitation in our understanding of it. It opens questions: is MoL actually learning to skip computations for easier tasks, or is the layer activation difference a byproduct of generating fewer tokens?

**Questions:**

1) MoL's adaptive strategy relies on identifying when an initial answer is incorrect (to trigger extended reasoning). During training, this is determined by checking the answer's F1 against ground truth. How is this handled at inference time, when the ground truth isn't available?

2) The current difficulty assessment is tailored to multi-document QA, using cross-document redundancy as a proxy. How would MoL extend to scenarios in single-document QA or tasks like mathematical problem solving (where all information is in one passage but reasoning is needed)?

3) MoL introduces several hyperparameters (for reward balancing, difficulty thresholds, target lengths, etc.). Did the authors observe any sensitivity or need for tuning these on different datasets?

4) The paper finds a correlation between shorter answers and fewer activated layers. Can the authors shed more light on this adaptive computation aspect? For instance, did they analyse whether specific transformer layers are being skipped or pruned dynamically, or is it more that fewer tokens cause fewer layers to effectively contribute?

5) Given the reliance on RL and considerable compute, how feasible is it to apply MoL to much larger models or other settings?

6) Do the authors anticipate any challenges scaling to 65B or GPT-3-sized models with this approach?

7) Are there any failure modes observed for MoL (cases where it chooses the wrong length adaptation)?

---

> ### Author Response · Authors · 2025-11-21
> **Response to Reviewer Wcic (1/2)**
>
> We thank the reviewer for the detailed review of the paper and the valuable feedback. Below, we address the reviewer's questions in a point-by-point manner.
> ***
> **W1 & Q2: Lack of applicability to single-document QA or other formats.**
>
> We agree with the reviewer that our current difficulty metric, based on cross-document redundancy, is specifically designed for multi-document scenarios and has limited direct applicability to tasks whose difficulty primarily stems from complex intra-document reasoning.
>
> Although our difficulty assessment method requires multi-document tasks as input, the proposed bidirectional reward function is task-agnostic and generalizes to diverse NLP tasks beyond multi-document settings. We would like to draw attention to the generalization results reported in Table 5. Performance on CommonsenseQA and SVAMP indicates that models trained with MoL still outperform the baseline on tasks that are not document-centric. This suggests that, although the heuristic may not be optimal in those settings, MoL helps the model acquire a more general and intrinsic ability to assess question difficulty and to allocate an appropriate computation budget. For tasks without multi document contexts a predefined set of initial target lengths derived from approximate output token lengths suffices to provide provisional guidance. The bidirectional reward function in MoL automatically identifies optimal response patterns through iterative reward optimization.
>
> We have added a discussion of these points in Section 7 (lines 523-530) to clarify MoL’s adaptability.
> ***
> **W2 & Q3: Complexity of tuning hyperparameters.**
>
> We acknowledge that our framework introduces several hyperparameters. In Appendix B, we provide comprehensive ablation studies across all hyperparameters to validate the robustness of our proposed approach (including four newly experiment: (1) Fixed initial length configurations , (2) k-value sensitivity analysis, (3) Embedding encoder , and (4) Sentence-level unit).
>
> Our main findings are as follows. Because our progressive learning strategy dynamically relaxes constraints, different initial target length settings yield comparable performance gains. The primary effect of these hyperparameters is on training time rather than on final task effectiveness. Moreover, key parameters such as α provide an intuitive tuning knob for the accuracy and efficiency tradeoff. Although identifying a globally optimal configuration can be complex, a broad range of parameter settings already produces substantial improvements over the baseline.
> ***
> **W3 & Q5, Q6: High computational cost, accessibility, and scalability.**
>
> We acknowledge the reviewer’s concern about computational cost. The use of 64 A100 GPUs reflects the demanding nature of state of the art reinforcement learning based alignment methods such as GRPO and is not intrinsic to MoL. Many of our experiments can be reproduced with 4 A100 GPUs and most require only 8 A100 GPUs, with larger resource requirements only for experiments involving the 14B model. We believe this one time training cost should be weighed against MoL’s primary benefit of reducing inference compute and latency. For large production models even modest reductions in average response length lead to substantial long term savings in compute and improved user experience. MoL is model independent and we expect its value to increase for models where efficiency is critical. The main practical challenge is the engineering and hardware required for the reinforcement learning training stage, which is comparable to other large scale fine tuning efforts.
>
> We have demonstrated through model generalization experiments that MoL scales effectively across different model sizes, consistently achieving strong performance. Due to limited computational resources, we are currently unable to conduct experiments on larger models; however, based on our existing results, we believe that scaling MoL to 65B or GPT-3-sized models is feasible.

---

> ### Author Response · Authors · 2025-11-21
> **Response to Reviewer Wcic (2/2)**
>
> **W4: Need for human evaluation for conciseness.**
>
> To address it, we conducted a human evaluation on 100 randomly sampled compressed outputs from the dataset. For each question, domain experts were shown both the compressed response produced by the MoL trained model and the corresponding response from the pretraining baseline, and were asked to assess readability and information sufficiency. They found that 94 percent of the compressed outputs were overall better than the pretraining baseline. We have added a selection of representative examples to Appendix F.
> ***
> **W5 & Q4: Correlational nature of layer activation analysis.**
>
> The reviewer’s observation is correct. Our analysis is indeed post hoc and correlational, and cannot definitively establish causality. The reviewer raises an important question: is MoL learning to skip computation, or are the observed differences in layer activations a byproduct of generating fewer tokens? We consider the former explanation more plausible. After training with MoL, the model exhibits a higher average number of activated layers when answering difficult questions compared to the pre-training baseline, while simultaneously producing more concise outputs (i.e., shorter token sequences). This observation contradicts the latter hypothesis, which attributes the activation differences purely to fewer generated tokens. Instead, we believe that, after training, the model learns to adaptively select layers for reasoning: for simple questions, only a small number of layers are needed, whereas more complex questions require deeper layer utilization.
> ***
> **Q1: Adaptive strategy for MoL at inference time**
>
> The dual objective reward mechanism $R_{extend}$ and $R_{compress}$ is solely a training time strategy intended to teach the model an adaptive policy. During the inference, the trained model is directly used.
> ***
> **Q7: Are there any failure modes observed for MoL?**
>
> This is an excellent question. We identify two failure modes in our difficulty heuristic. First, when information across documents is highly similar but solving the question requires deep multi step reasoning, the heuristic can underestimate difficulty and label the question as easy. Second, when documents are highly divergent yet the answer can be directly extracted from a single document, the heuristic can overestimate difficulty and label the question as hard. We add a focused analysis of these two patterns in Appendix F.3.
>
> Importantly, incorrect difficulty labels do not necessarily produce wrong final answers but they do affect training efficiency (Figure 7 & Figure 8). Our dual reward scheme based on answer correctness supplies a self correcting pressure during training. For example, if a hard question is misclassified as easy, the model may produce a short but incorrect answer and then receive a penalty from the extend reward, which encourages the model to explore longer responses and ultimately find the correct answer.
> ***
> Thank you again for your insightful feedback, that helped us improve our work.

---

### Official Review · Reviewer_5HsN · 2025-11-07

**Soundness:** 2
**Presentation:** 3
**Contribution:** 2
**Rating:** 2
**Confidence:** 4

**Summary:**

This paper introduces Mixture-of-Length (MoL), a novel framework for question answering (QA) with context that aims to improve the balance between reasoning quality and computational efficiency. The authors identify a tension in current methods, which either compress responses uniformly (harming complex reasoning) or rely on simple heuristics for adaptive reasoning.

The authors demonstrate that this training leads to an "intelligent brevity" behavior, where the model naturally uses shorter responses for simple queries and longer ones for complex queries. Experiments on HotpotQA, StrategyQA, and Loong show that MoL-trained models achieve higher accuracy while simultaneously reducing the number of generated tokens compared to baselines . A post-hoc analysis also suggests this adaptive output length correlates with adaptive internal computation, where easy questions activate fewer transformer layers .

**Strengths:**

The paper tackles a significant and well-recognized problem: the trade-off between reasoning quality and efficiency (cost/latency) in LLMs. The goal of achieving "intelligent brevity" is highly desirable for real-world applications.

**Weaknesses:**

*The "Principled" Difficulty Metric Seems Oversold*

The paper's first contribution, the "principled difficulty assessment" is framed with a sophisticated theoretical grounding in information theory and the NP-hard Set Cover problem. However, the practical implementation is a simple heuristic: the average inter-document cosine similarity of k extracted sentences. This feels like a significant oversell: what is the **information theory** used? Under **which optimization** you derive the metric? What is the relationship between the NP-hard and the proposed difficulty calculation?


*The Difficulty Heuristic Is Potentially Flawed*

The core assumption that high similarity (redundancy) equals "easy" and low similarity equals "hard" is questionable and may not hold in many cases.

(Hard, High-Sim): A question might require synthesizing **subtle differences** between two **very similar documents**. The high similarity would incorrectly classify this as "easy".

(Easy, Low-Sim): A question might be a simple fact-retrieval task where the two facts just happen to reside in two completely dissimilar documents. The low similarity would incorrectly classify this as "hard".

*The Effectiveness Proof of Reward Mechanism*

The paper's core innovation is presented as the dual-reward mechanism (compress-if-correct, extend-if-wrong). However, this mechanism is entangled with a second mechanism: the dynamic L_target . The paper's claim of "emergent brevity" is muddled because it's impossible to know if the model is adapting because of **the dual reward** or simply because it's being **explicitly told to use a different target length**.

If the reward system can be truely useful to learn to reward or punish on the length, it is highly recommend to add a study to set the fix length for all difficulties, and check the performances.

**Questions:**

See weakness

---

> ### Author Response · Authors · 2025-11-21
> **Response to Reviewer 5HsN (1/2)**
>
> We thank the reviewer for the detailed review of the paper and the valuable feedback. Below, we address the reviewer's questions in a point-by-point manner.
> ***
> > The "Principled" Difficulty Metric Seems Oversold
> >
> > The paper's first contribution, the "principled difficulty assessment" is framed with a sophisticated theoretical grounding in information theory and the NP-hard Set Cover problem. However, the practical implementation is a simple heuristic: the average inter-document cosine similarity of k extracted sentences. This feels like a significant oversell: what is the information theory used? Under which optimization you derive the metric? What is the relationship between the NP-hard and the proposed difficulty calculation?
>
> We agree that our original wording overstated the formal mathematical connection between our practical metric and the information‑theoretic literature. Below we summarize how we will clarify and strengthen the paper in response.
>
> From an information‑theoretic perspective:
>
> - We model QA as needing to assemble a set $U$ of key information snippets, where each document $D_i$ provides a subset $A_i$ ⊆ $U$. The intrinsic problem difficulty is intuitively related to the size of the minimal set cover: fewer distinct documents required to cover $U$ implies higher redundancy and lower reasoning complexity. This modeling provides motivation and intuition rather than a formal equivalence.
> - Higher redundancy corresponds to a lower minimal description length, reducing the rate‑distortion cost of producing a correct answer. We will rephrase these links as motivating intuitions rather than formal theorems.
>
> Why we use average inter‑document similarity as a practical proxy and its relation to NP‑hardness?
>
> * Exact computation of the minimal set cover is NP‑hard and infeasible at large scale. Therefore we adopt a computationally tractable heuristic: extract Top‑k question‑relevant sentences from each document (to reduce noise), then compute the average pairwise cosine similarity S̄ between these filtered texts. Intuitively, higher S̄ indicates greater cross‑document overlap and thus a smaller effective cover size; we set difficulty as C = 1 − S̄.
> * We do not claim S̄ is mathematically equivalent to the minimal set cover. Rather, S̄ is a practically computable proxy motivated by the set‑cover intuition. We will explicitly weaken the “principled” language and clarify that our work follows a “theoretical motivation → computable approximation” path.
>
> Following your suggestion, we have merged the aforementioned analysis into Section 3.2(lines 153-163,184-189) in the revised manuscript. Please refer to the updated version for details.
> ***
> > The Difficulty Heuristic Is Potentially Flawed
> >
> > The core assumption that high similarity (redundancy) equals "easy" and low similarity equals "hard" is questionable and may not hold in many cases.
> >
> > (Hard, High-Sim): A question might require synthesizing subtle differences between two very similar documents. The high similarity would incorrectly classify this as "easy".
> >
> > (Easy, Low-Sim): A question might be a simple fact-retrieval task where the two facts just happen to reside in two completely dissimilar documents. The low similarity would incorrectly classify this as "hard".
>
> We thank the reviewer for this insightful counterexample. We fully acknowledge that a similarity-based heuristic is not infallible and can fail in the scenarios you describe. At the same time, we would like to gently point out that the empirical results in Table 4 and Figure 3 show our heuristic to be effective in the majority of cases: model performance aligns with our difficulty scores (higher accuracy on simple problems), indicating that, while not universally perfect, the metric is a practical proxy for the type of multi-document QA tasks we study.
>
> We further emphasize that MoL’s core strength lies in its adaptive modulation of reasoning strategies. Even when the initial difficulty assessment is inaccurate, the model gradually rectifies errors through the dual-objective reward mechanism during training. Specifically, incorrect initial predictions incur higher penalties , prompting the model to dynamically allocate computational resources toward more accurate solutions. As training progresses, the interplay between compression and extension rewards enables the model to self-correct and converge toward optimal response strategies. This behavior is empirically validated in Appendix F.3, where we visualize the evolution of outputs for challenging and straightforward questions at different training stages, demonstrating the model’s progressive alignment with ground truth difficulty levels.
>
> Thank you again for the suggestion; it points to valuable directions for improving difficulty estimation in future work.

---

> ### Author Response · Authors · 2025-11-21
> **Response to Reviewer 5HsN (2/2)**
>
> >The Effectiveness Proof of Reward Mechanism
> >
> >The paper's core innovation is presented as the dual-reward mechanism (compress-if-correct, extend-if-wrong). However, this mechanism is entangled with a second mechanism: the dynamic $L_{target}$ . The paper's claim of "emergent brevity" is muddled because it's impossible to know if the model is adapting because of the dual reward or simply because it's being explicitly told to use a different target length.
> >
> >If the reward system can be truely useful to learn to reward or punish on the length, it is highly recommend to add a study to set the fix length for all difficulties, and check the performances.
>
> Thank you for this valuable suggestion. To validate it, we ran controlled experiments with the Qwen3-8B model, fixing the target response length to 256, 512, and 1024 tokens and evaluating performance across our datasets. The results are as follows:
> |Method|HotpotQA (Acc)|HotpotQA (Tokens)|StrategyQA (Acc)|StrategyQA (Tokens)|Loong (Acc)|Loong (Tokens)|
> |:--------|:--------|:--------|:--------|:--------|:--------|:--------|
> |Original|61.0|609|93.7|468|55.8|2165|
> |MoL(256,512,1024)|66.7|303|96.1|247|61.4|1473|
> |MoL(512,1024,2048)|67.2|316|95.9|219|**62.3**|1374|
> |MoL(256,256,256)|67.0|303|**96.4**|236|61.7|**1338**|
> |MoL(512,512,512)|**67.4**|331|95.9|**213**|62.1|1402|
> |MoL(1024,1024,1024)|66.9|**286**|96.2|225|61.3|1425|
>
> The results show that final performance is largely insensitive to the initial length setting: accuracy remain comparable across the three fixed-length settings. However, analysis of training dynamics reveals a clear effect on convergence speed: when the initial length is closer to the optimal response length, the model reaches convergence in fewer training steps (For detailed information, see Figure 7 and Figure 8 in Appendix B.5).
>
> We therefore conclude that the chosen target length primarily impacts training efficiency rather than the ultimate effectiveness of the trained model. This suggests practitioners can select an approximate target length to accelerate training without materially affecting final performance. Based on the experimental results, different target length selctions have marginal influence to the overall performance.
> ***
> Thank you again for your insightful feedback, that helped us improve our work.

---

### Author Response · Authors · 2025-11-21
**Summary of Paper Revision**

We thank the reviewers for their detailed evaluation of our manuscript and their constructive feedback. Based on the recurring questions and suggestions, we have revised the manuscript, with all changes marked in blue in the updated draft. Below we summarize the main changes and additions.

- Section 3.2 has been reorganized to better clarify the relationship between theory and practice.
- Appendix B.5: We added experiments with three additional values of $L_{target}$, as well as new experiments that investigate the impact of $L_{target}$ on training efficiency.
- New Appendix B.7: We added ablation studies on the choice of k in Top k sentence filtering, the embedding model, and the sentence granularity.
- New Appendices F.3 and F.4: We added illustrative examples of incorrect difficulty estimation and failure cases of MoL, respectively.
- We added a discussion of the limitations in the limitations section.

We hope these updates adequately address the reviewers concerns and improve the clarity, generality, and relevance of our work.
We are happy to make further adjustments or provide additional clarifications during the remaining discussion period.

---

### Author Response · Authors · 2025-11-27

Dear Reviewers,

I hope this message finds you well. As the discussion phase is nearing its end, we would like to kindly check whether our previous responses have addressed your concerns. We totally understand that this is quite a busy period, and we sincerely appreciate the time and effort you have devoted to reviewing our work. If there are any remaining questions, we would be more than happy to clarify them to the best of our ability.

Best regards,

The Authors

---

### Author Response · Authors · 2025-12-01
**Summary to AC (1/2)**

Dear Area Chair,

We sincerely thank all reviewers and the area chair for their valuable comments and constructive feedback on our paper. Given that most reviewers did not respond during the discussion phase, we sincerely apologize for adding to the area chair’s reviewing burden. To help reduce area chair reviewing burden, we have merged several related concerns raised by different reviewers and provide a summary below, so that you can more quickly grasp the main points.

**Method overview**

Our paper proposes Mixture of Length (MoL), an adaptive reasoning framework for context based question answering. MoL combines a difficulty estimation with a dual objective reward mechanism, enabling the model to automatically adjust its reasoning length according to question difficulty: it produces more concise answers for simple questions and proactively extends the reasoning chain for complex ones, thereby achieving intelligent brevity that balances "reasoning quality" and "computational efficiency." We conduct systematic experiments on multiple datasets and base models. MoL consistently improves accuracy while significantly reducing the number of output tokens, demonstrating its potential for resource constrained LLM applications.

**Summary of rebuttal process and revisions:**

>Reliability of the difficulty estimation method

To address the concern "Is the difficulty partition reliable", we strengthened the evaluation from two angles:

1) Comparison with large model prompt based difficulty assessment: we use DeepSeek V3 to perform prompt based difficulty annotation and compare model accuracy on the easy and hard subsets under different partitioning methods. The results show that, when using the MoL proxy indicator for partitioning, accuracy on simple questions is significantly higher and accuracy on difficult questions is significantly lower, which better matches the intuitive difficulty distribution.

2) Systematic ablations and robustness analysis: in Appendix B.7, we add multiple experiments to test the sensitivity of difficulty assessment to key design choices, including:
   - Top k sentence selection
   - Embedding model
   - Sentence granularity

The results show that performance remains stably improved under different settings, indicating that MoL is robust to these design choices.

In addition, we add concrete examples of difficulty misestimation and MoL failure cases in Appendix F.3 and F.4, respectively, to more transparently illustrate the behavior and limitations of the method on challenging instances and to show that the model can still exhibit a self correction tendency in some cases.

>Source of performance gains: dual reward mechanism vs. target length setting

To address this question, we add two key sets of experiments:

1) Fixed global target length: we fix $L_{target}$ to 256, 512, and 1024 and train with the same target length for all samples. The results show that even without difficulty dependent target lengths, MoL still achieves clearly better performance than GRPO and the original model, indicating that the core benefit comes from the dual reward mechanism rather than from the length stratification itself.

2) Evolution of length during training: we track the changes of output length and accuracy across epochs (Figure 7 and 8) and find that the main effect of the target length is on the speed and stability of length adaptation during training, while its impact on the final converged performance is minor. This further supports the conclusion that the adaptive behavior is driven by the dual reward, rather than solely by the initial length setting.


>Relationship between difficulty estimation, information theory, and set cover

We apologize for the confusion caused by our theoretical motivation in Section 3.2. We have reorganized and clarified this section: we now explicitly state that our difficulty estimator is a "computable proxy" inspired by the set cover problem, used to characterize the relationship between cross document redundancy and reasoning difficulty, rather than a formal equivalence or exact solution to the original problem. This more clearly separates theoretical motivation from practical implementation.

>Hyperparameter tuning

In Appendix B, we provide comprehensive ablation studies for all hyperparameters to verify the robustness of our method, including four new experiments: (1) fixed initial length configuration, (2) sensitivity analysis of k, (3) choice of embedding encoder, and (4) sentence level units. The results show that within a broad parameter range, MoL consistently outperforms the original model and RL baselines, indicating that the method is fairly robust to hyperparameter changes and does not require fine grained tuning to deliver significant gains in practice.

---

> ### Author Response · Authors · 2025-12-01
> **Summary to AC (2/2)**
>
> > Extension to single document or non document tasks
>
> Some reviewers were concerned that MoL might be limited to multi document scenarios. In the rebuttal, we emphasized and empirically validated that the core reward mechanism of MoL is agnostic to task type and input format. To this end, we conduct experiments on CommonsenseQA and SVAMP (Table 5). The results show that even in commonsense reasoning and math word problem tasks without explicit document retrieval, MoL can still control length while improving accuracy. Together with the experiments in Appendix B, which show that the target length mainly affects training efficiency rather than final performance, we argue that in single document or non document settings one only needs to set a reasonable initial $L_{target}$ based on the typical output length of the task, and MoL remains applicable. We further discuss this point in the LIMITATIONS section.
>
> > Readability of the trained model’s outputs
>
> Reviewers were concerned that compressed outputs might become hard to read or unnatural. To provide stronger evidence, we conduct a human evaluation on 100 samples, comparing readability and information sufficiency between MoL outputs and baseline outputs. The results show that in 94 percent of the samples, annotators prefer the compressed outputs from MoL, judging them to be more concise and focused without harming understanding. At the same time, we add failure cases in Appendix F.4 and show behaviors under difficulty misclassification in Appendix F.3, to illustrate in what situations over compression or information omission may occur and thus delineate the applicability boundaries of MoL.
>
> >Whether changes in activated layers are merely a byproduct of shorter outputs
>
> We compare model activation patterns before and after MoL training in the rebuttal. The key observation is that on difficult questions, the model trained with MoL exhibits two simultaneous changes relative to the baseline:
> 1) A higher average number of activated layers, that is, deeper layers are more effectively engaged in reasoning.
> 2) Shorter and more concise outputs than the baseline.
>
> This contradicts the hypothesis that changes in activated layers are passively caused by shorter outputs. Instead, it suggests that the model has learned to adaptively allocate internal computation resources according to difficulty: using fewer layers and shorter outputs for simple questions, and engaging deeper reasoning with more layers for complex questions while avoiding unnecessary verbosity. Figure 4 provides a more detailed visualization of this effect, but we also candidly note in the LIMITATIONS section that this relationship is based on correlational analysis and that more systematic causal and mechanistic studies are needed in future work.
>
> > Further clarifying the correlations among difficulty, length, and accuracy
>
> To more rigorously support the link between the difficulty proxy and model behavior, we add a Spearman correlation analysis on HotpotQA: the correlation between similarity and accuracy is ρ = 0.91, while that between similarity and length is ρ = −0.93, statistically confirming that our similarity based proxy is highly correlated with actual task difficulty and model output patterns. This provides additional evidence for the information theoretic motivation behind our difficulty design.
>
> > Clarifying figures, notation, and statements
>
> In response to the reviewers’ presentation related comments, we carefully revised the manuscript:
>
> We clarify Figure 2 to explicitly indicate that "compression" and "expansion" are modes dynamically switched by the reward function, rather than two fixed phases in training. We adjust the font sizes and layout of Figure 5 to improve readability. We consolidate the definitions of all symbols in Section 3.2, unify the notational conventions, and clearly state in the main text that "tokens" refer only to output tokens. These revisions aim to reduce reading burden and make the design and experiments of MoL more intuitive.
>
> We sincerely appreciate the reviewers for their rigorous and insightful comments and questions, which helped us refine the theoretical exposition, experimental design, and presentation, and pushed us to more thoroughly examine and demonstrate the effectiveness and limitations of MoL. We also thank the area chairs for coordinating and facilitating the discussion. We have submitted a revised version, and we believe that the current manuscript has been substantially improved in clarity, rigor, and persuasiveness.
>
> Sincerely,
>
> The authors

---

### Meta-Review · Area_Chair_oW3J · 2026-01-09

**Summary:**

Most reviewers provided positive evaluations, and the author rebuttal has adequately addressed the major concerns raised. While the work still has some limitations, e.g., the reliance on hand-tuned thresholds that may not generalize well to new setting, these issues do not constitute fundamental flaws. Overall, the paper is technically sound, and the newly added experiments and human studies provide solid support for the claimed contributions. Therefore, I believe the paper merits acceptance.

**Reviewer Concerns:**

R1’s concerns are not fully resolved; however, the rebuttal substantially reduces their severity.

**Reviewer Scores:**

Although R1 has not responded to the rebuttal and may not revise their score, I believe the rebuttal justifies an increase from a score of 2. If such an increase were made, this paper would become the highest-rated submission in my batch.

---

### Decision · Program_Chairs · 2026-01-26

Accept (Poster)